# Subcellular proteomics of dopamine neurons in the mouse brain

Benjamin D Hobson[1,2,3], Se Joon Choi[4], Eugene V Mosharov[4], Rajesh K Soni[5], David Sulzer[3,4,6,7,8]*, Peter A Sims[1,8,9,10]*

[1]Department of Systems Biology, Columbia University Irving Medical Center, New York, United States; [2]Medical Scientist Training Program, Columbia University Irving Medical Center, New York, United States; [3]Department of Psychiatry, Columbia University Irving Medical Center, New York, United States; [4]Division of Molecular Therapeutics, New York State Psychiatric Institute, New York, United States; [5]Proteomics Shared Resource, Herbert Irving Comprehensive Cancer Center, Columbia University Irving Medical Center, New York, United States; [6]Department of Neurology, Columbia University Irving Medical Center, New York, United States; [7]Department of Pharmacology, Columbia University Irving Medical Center, New York, United States; [8]Aligning Science Across Parkinson's (ASAP) Collaborative Research Network, Chevy Chase, United States; [9]Department of Biochemistry & Molecular Biophysics, Columbia University Irving Medical Center, New York, United States; [10]Sulzberger Columbia Genome Center, Columbia University Irving Medical Center, New York, United States

*For correspondence:
ds43@cumc.columbia.edu (DS);
pas2182@columbia.edu (PAS)

**Competing interest:** The authors declare that no competing interests exist.

**Abstract** Dopaminergic neurons modulate neural circuits and behaviors via dopamine (DA) release from expansive, long range axonal projections. The elaborate cytoarchitecture of these neurons is embedded within complex brain tissue, making it difficult to access the neuronal proteome using conventional methods. Here, we demonstrate APEX2 proximity labeling within genetically targeted neurons in the mouse brain, enabling subcellular proteomics with cell-type specificity. By combining APEX2 biotinylation with mass spectrometry, we mapped the somatodendritic and axonal proteomes of midbrain dopaminergic neurons. Our dataset reveals the proteomic architecture underlying proteostasis, axonal metabolism, and neurotransmission in these neurons. We find that most proteins encoded by DA neuron-enriched genes are localized within striatal dopaminergic axons, including ion channels with previously undescribed axonal localization. These proteomic datasets provide a resource for neuronal cell biology, and this approach can be readily adapted for study of other neural cell types.

## Editor's evaluation

In this work, the authors provide a useful compendium of proteins labeled within dopaminergic cells using a novel approach. Novel viral approaches were developed to rapidly biotinylate proteins in dopaminergic neurons in oriented sections of brain whereby circuits can be spatially parsed for proteomic dissection. In addition to providing a useful new database of proteins for investigators interested in this circuit, the results also provide a more general approach to examining a compartment proteome in neurons and what might be expected in that analysis in an unbiased way not previously envisaged.

## Introduction

Dopamine (DA) release from the axons of midbrain dopaminergic (mDA) neurons provides important signals that regulate learning, motivation, and behavior. Given that dopaminergic dysfunction is linked to neuropsychiatric diseases including Parkinson's disease (PD), schizophrenia, and drug addiction,

there is considerable interest in deep molecular profiling of mDA neurons in health and disease. Molecular profiling of striatal DA axons is of particular interest, since they may be the initial site of mDA neuronal degeneration in PD (*Burke and O'Malley, 2013*), a point strongly supported by neuro-pathological comparison of dopaminergic axonal and cell body loss in PD patients (*Kordower et al., 2013*). Thus, the study of proteins in striatal mDA axons and how they are altered in disease or disease models is a topic of intense investigation.

Despite their importance in behavior and disease, mDA neurons are very few in number, with an estimated ~21,000 in the midbrain of C57BL/6 mice (*Nelson et al., 1996*). Thus, although proteomic profiling of midbrain tissue samples offers insight into PD pathophysiology (*Jung et al., 2017*; *Petyuk et al., 2021*), such studies do not analyze striatal mDA axons, nor can they identify the cellular source of changes in protein levels. Most mDA neuron-specific molecular profiling studies have focused on mRNA using translating ribosome affinity purification (TRAP) or single-cell RNA-sequencing (scRNA-seq) (*Agarwal et al., 2020*; *Brichta et al., 2015*; *Dougherty, 2017*; *Kramer et al., 2018*; *Poulin et al., 2014*; *Saunders et al., 2018*; *Tiklová et al., 2019*). Although these methods have advanced our understanding of gene expression in mDA neurons, there are important issues that are not addressed by the study of mRNA. First, mRNA levels do not always correlate with protein abundance, particularly for axonal proteins (*Moritz et al., 2019*). Second, transcriptomics cannot establish the localization or abundance of the encoded proteins within specific subcellular compartments, which is particularly important for mDA neurons.

After exiting the midbrain, the axons of mDA neurons travel within the medial forebrain bundle (MFB) to innervate forebrain structures. mDA axonal arbors within the striatum are immense and highly complex: single mDA neuron tracing has shown that mDA axons can reach over 500,000 μm in total length (*Matsuda et al., 2009*). Although their axons constitute the major volume and energy demand for mDA neurons (*Bolam and Pissadaki, 2012*; *Matsuda et al., 2009*; *Pacelli et al., 2015*), dopaminergic axons represent only a small fraction of tissue protein in the striatum. The ability to directly interrogate subcellular proteomes of mDA neurons in native brain tissue would significantly enhance our understanding of mDA neuronal biology.

To study the subcellular proteome of mDA neurons in the mouse brain, we have adapted enzyme-catalyzed proximity labeling combined with mass spectrometry (MS)-based proteomics, a powerful approach for identifying protein interactions and/or localization in subcellular compartments (*Hung et al., 2014*; *Loh et al., 2016*; *Rhee et al., 2013*; *Roux et al., 2012*). For in-cell proximity labeling, a particularly efficient enzymatic approach uses the engineered ascorbate peroxidase APEX2 (*Lam et al., 2015*). APEX2 rapidly biotinylates proximal proteins in the presence of hydrogen peroxide ($H_2O_2$) and biotin-phenol (BP), generally requiring <1 min of labeling in cell culture (*Hung et al., 2016*; *Hung et al., 2017*). While most proximity labeling studies have been conducted on cultured cells, APEX2 proximity labeling has also been demonstrated in live *Drosophila* tissues (*Chen et al., 2015*), and recently in mouse heart (*Liu et al., 2020*). Selective expression of APEX2 enabled electron microscopy reconstructions of genetically targeted neurons in mice (*Joesch et al., 2016*; *Zhang et al., 2019*), suggesting that APEX2 may be suitable for cell type-specific proximity labeling and proteomics in live brain tissue.

Here, we employ APEX2-mediated biotin labeling in acute brain slices to study the subcellular proteome of mDA neurons. Using a combination of cell type-specific APEX2 expression and proximity biotinylation in acutely prepared slices, we have characterized the somatodendritic and axonal proteomes of midbrain mDA neurons. We show that the striatal axonal arbors contain nearly 90% of mDA neuronal proteins accessible to cytoplasmic APEX2 labeling, providing robust coverage of proteins involved in axonal transport, DA transmission, and axonal metabolism. Of particular interest, we find that many proteins encoded by mDA neuron-enriched genes are preferentially localized in striatal axons. Together, our data establish a proteomic architecture for mDA neurons and identify candidates for mechanistic follow-up studies of PD-relevant mDA neuronal cell biology.

## Results

### APEX2 enables rapid, mDA neuron-specific biotin labeling in acute brain slices

To express cytoplasmic APEX2 specifically in mDA neurons, we employed an AAV5 viral vector containing the Cre-dependent construct developed by *Joesch et al., 2016*, which expresses a V5-tagged APEX2 fused to a nuclear export sequence (NES) (*Wen et al., 1995*). We injected AAV5-CAG-DIO-APEX2NES into the ventral midbrain (VM) of DAT-IRES-Cre mice (*Bäckman et al., 2006*; *Figure 1a*). To confirm the cellular specificity and Cre dependence of APEX2 expression, we injected the virus into DAT-IRES-Cre mice crossed with Ai9 tdTomato reporter mice (*Madisen et al., 2010*). Immunostaining against V5 (APEX2), RFP (tdTomato), and tyrosine hydroxylase (TH), a canonical mDA neuron marker, demonstrated specific expression of APEX2 in mDA neurons (*Figure 1b*). We found that ~97% of V5-APEX2$^+$ neurons were also double-positive for tdTomato and TH, while V5-APEX2 expression was never detected in nondopaminergic (TH$^-$/tdTomato$^-$) neurons (*Figure 1—figure supplement 1a, b*). All three markers displayed intense staining throughout the mDA neuronal cytoplasm, including dendrites in the VM and axonal projections in the striatum (*Figure 1b*). As determined by quantification of fluorescence intensity in confocal images, the subcellular distribution of V5-APEX2 concentration was indistinguishable from TH and tdTomato (*Figure 1—figure supplement 1c, d*). The average intensity for all three proteins was slightly higher in mDA neuronal somata than in dendrites and axons (*Figure 1—figure supplement 1c, d*), which may reflect somatic protein synthesis and outward transit into dendrites and axons. Importantly, these data show no bias in the distribution of V5-APEX2 concentration compared to other cytoplasmic proteins in mDA neurons. Thus, injection of Cre-dependent AAV-APEX2NES (hereafter referred to as APEX2) into the VM of DAT-IRES-Cre mice leads to robust expression of APEX2 throughout the mDA neuronal cytoplasm.

As shown in *Drosophila* (*Chen et al., 2015*), APEX2 labeling can be performed in living tissue in the presence of $H_2O_2$ and BP. We conducted APEX2 labeling in acute brain slice preparations under conditions that preserve the integrity of neurons and severed axons, as shown by electrophysiological recordings and stable levels of evoked DA release for at least 6 hr (*Hernandez et al., 2012*). Typical sets of coronal or sagittal slices are shown in *Figure 1—figure supplement 2a* and our workflow is summarized in *Figure 1c*. Key steps in the protocol include: (1) transcardial perfusion with low-sodium cutting solution, which acts to preserve neuronal integrity in slices from adult mice (*Ting et al., 2014*) and removes catalase-rich blood, (2) incubation of slices with BP in oxygenated artificial cerebrospinal fluid (aCSF) during the slice recovery period, (3) rapid labeling with $H_2O_2$ in aCSF, and (4) rapid quenching by transferring slices to antioxidant aCSF. Because the slices are far thicker (300 μm) than monolayer cell cultures, we fixed, cleared, and stained them with fluorescent anti-V5 (APEX2) and streptavidin after biotin labeling in 1 mM $H_2O_2$ for 1–5 min (*Figure 1—figure supplement 2b*). We found that biotinylation was detectable at all time points, although streptavidin labeling appeared weaker within the center of slices, suggesting incomplete penetration of BP and/or $H_2O_2$. Rather than targeting a specific organelle or protein complex, our goal in this work was to broadly label the entire cytoplasm of mDA neurons. Therefore, we chose 3 min of $H_2O_2$ exposure for downstream applications, which provided sufficient labeling for proteomics while limiting $H_2O_2$ exposure.

Western blotting of slices treated with BP and $H_2O_2$ showed broad biotinylation patterns in the midbrain and striatum (*Figure 1d*), consistent with labeling of somatodendritic and axonal proteins, respectively. Fluorescent streptavidin staining of sagittal slices after labeling and fixation revealed mDA neuron-specific labeling throughout the VM, MFB, and striatum (*Figure 1e*). We confirmed that both BP and $H_2O_2$ are required for APEX2-mediated biotinylation in mDA neuronal soma/dendrites and axons (*Figure 1f, g*). Confocal imaging of striatal slices revealed a dense, intricate staining pattern consistent with the cytoarchitecture of mDA axons (*Figure 1g*). Biotin labeling colocalized with V5-APEX2$^+$ mDA axons but not with surrounding somata or myelin tracts, since APEX-generated BP radicals do not cross membranes (*Rhee et al., 2013*). These results show that APEX2 can rapidly and specifically label the somatodendritic and axonal compartments of mDA neurons in acute brain slices.

### Proteomic profiling of subcellular compartments in mDA neurons

We next used mDA neuron-specific biotin labeling in slices to perform proteomic profiling of these neurons with subcellular resolution. After biotin labeling and quenching, we rapidly dissected and froze the VM, MFB, and striatum of sagittal slices for downstream biotinylated protein enrichment and

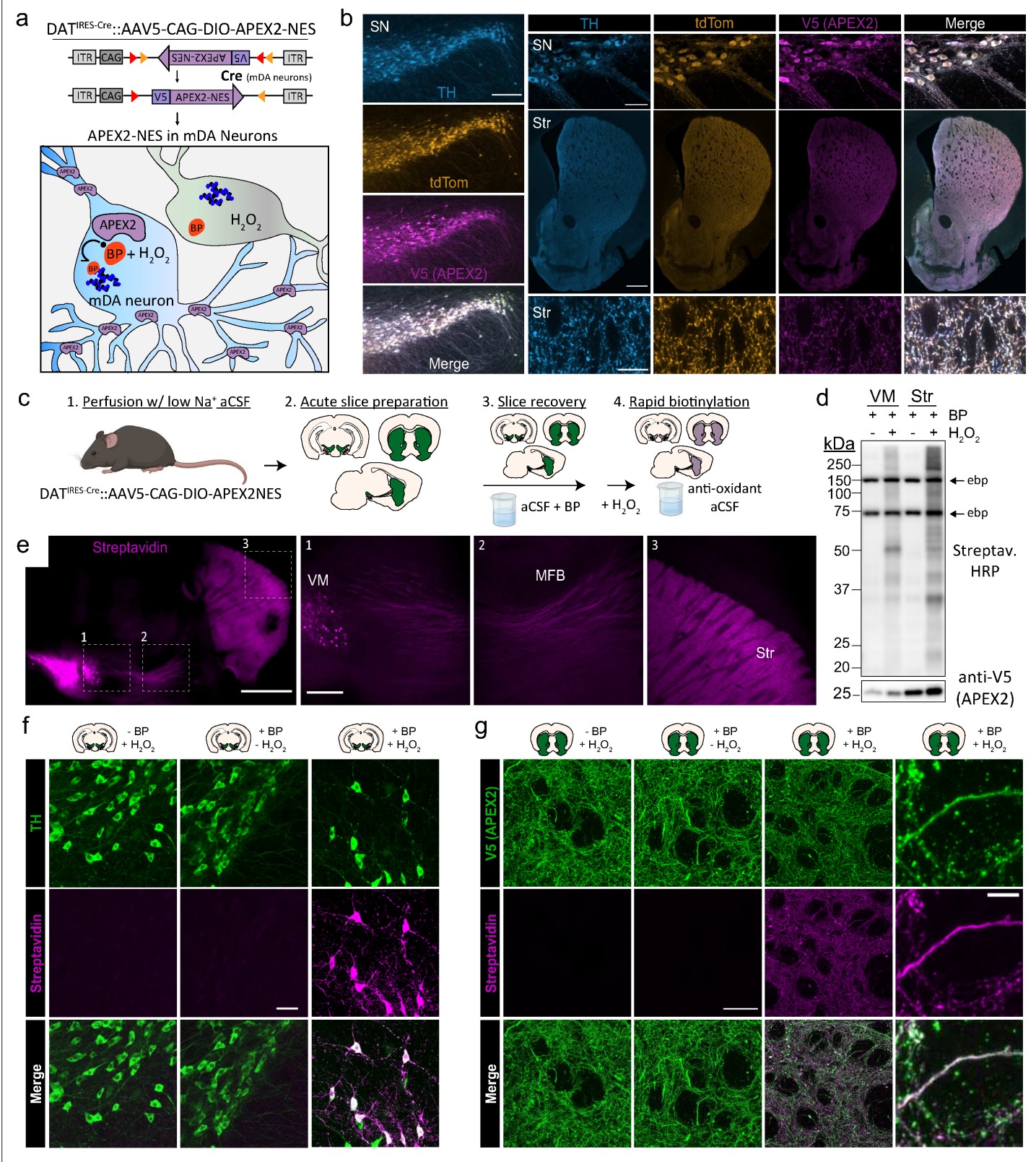

**Figure 1.** Cre-dependent viral expression of cytoplasmic APEX2 and midbrain dopaminergic (mDA) neuron-specific biotinylation in acute brain slices. (**a**) Schematic depicting viral expression strategy: Cre-dependent, cytoplasmic APEX2 expressing AAV (AAV5-CAG-DIO-APEX2-NES) is injected into the midbrain of DAT-IRES-Cre mice for mDA neuron-specific APEX2 labeling. (**b**) Immunostaining of mDA neurons in DAT-IRES-Cre/Ai9tdTomato mice injected with AAV5-CAG-DIO-APEX2-NES. Anti-TH, anti-RFP (tdTomato), and anti-V5 (APEX2) all display diffuse localization throughout somatic, dendritic, and

*Figure 1 continued on next page*

*Figure 1 continued*

axonal cytoplasm. *Left*, substantia nigra, scale bar: 200 μm, *upper right*: substantia nigra at high power, scale bar: 50 μm, *middle right*: dorsal and ventral striatum, scale bar: 500 μm, *lower right*: dorsal striatum at high power, scale bar: 10 μm. (**c**) Schematic depicting APEX2 labeling procedure in acute brain slices. Slice are incubated for 1 hr with 0.5 mM biotin phenol prior to labeling with 1 mM hydrogen peroxide ($H_2O_2$) for 3 min. (**d**) *Upper*: western blotting of ventral midbrain and striatal slice lysates with streptavidin–horseradish peroxidase in the presence of biotin phenol with or without $H_2O_2$. Endogenously biotinylated proteins (ebp) are noted in all lanes at ~75 and ~150 kDa. *Lower*: same as above but with anti-V5 (APEX2). (**e**) Streptavidin-AlexaFluor647 staining of sagittal slices after APEX2 labeling in mDA neurons. *Left*, sagittal slice at low power, scale bar: 1 mm. Insets indicated in white dashed lines, *right*: insets of the ventral midbrain, medial forebrain bundle, and striatum, scale bar: 250 μm. (**f**) Streptavidin-AlexaFluor647 staining of coronal midbrain slices after APEX2 labeling in mDA neurons, scale bar: 50 μm. Labeling requires both biotin phenol and $H_2O_2$. (**g**) Same as (**f**) but with fields of striatal slices. First three columns, scale bar: 50 μm. Far right panels at high magnification, scale bar: 5 μm. *Abbreviations*: aCSF, artificial cerebrospinal fluid; BP, biotin phenol; HRP, horseradish peroxidase; MFB, medial forebrain bundle; NES, nuclear export sequence; TH, tyrosine hydroxylase; SN, substantia nigra; Str, striatum; VM, ventral midbrain.

The online version of this article includes the following source data and figure supplement(s) for figure 1:

**Source data 1.** Western blots related to *Figure 1d*.

**Figure supplement 1.** Specificity of APEX2-NES AAV expression.

**Figure supplement 2.** Characterization of slice labeling.

liquid chromatography tandem mass spectrometry (LC–MS/MS) (*Figure 2a*). After lysis and protein precipitation to remove free biotin, we purified proteins from each region with streptavidin beads (streptavidin pulldown). To control for nonspecific binding and potential labeling by endogenous tissue peroxidases, we prepared slices from DAT-IRES-Cre mice without APEX2 (APEX2⁻, no virus control) and treated them identically to APEX2⁺ slices at all stages of the protocol. Streptavidin–horseradish peroxidase (HRP) blotting of captured proteins showed endogenously biotinylated carboxylase proteins at ~75 and ~150 kDa in all samples, while biotinylated proteins across a wide range of molecular weights were found only in APEX2⁺ samples (*Figure 2b*). Quantification of streptavidin–HRP signal revealed that approximately 87% of mDA neuronal APEX2 biotinylation is found within the striatum, 9% in the VM, and 4% in the MFB (*Figure 2d*). These results demonstrate that the majority of mDA neuronal proteins are found within striatal axons.

After on-bead tryptic digestion, we conducted label-free quantitative proteomics for single-mouse biological replicates of VM, MFB, and striatum streptavidin pulldown samples. Using data-independent acquisition (DIA), we quantified between 15,000 and 30,000 peptides representing 2100–2600 proteins per APEX2⁺ sample, while only ~5000 peptides representing ~1000 proteins were detected in APEX2⁻ samples (*Figure 2c*). Using the same LC–MS/MS workflow to analyze the bulk tissue proteome of VM and striatum slices, we found that approximately 45% of proteins quantified in the bulk tissue samples were also detected in APEX2⁺ samples (*Figure 2—figure supplement 1a*). More than 84% of proteins in both bulk tissue and APEX2⁺ streptavidin pulldown samples were identified based on quantification of multiple peptides (*Figure 2—figure supplement 1b*), and protein abundances of biological replicate APEX2⁺ streptavidin pulldown samples were highly correlated (*Figure 2—figure supplement 1c*; Pearson's $r = 0.93$–$0.94$ for striatum, $r = 0.89$–$0.92$ for VM, and $r = 0.82$–$0.84$ for MFB). Thus, even for specific subcellular compartments of small cell populations (~21,000 mDA neurons), the high efficiency of APEX2 labeling enables highly reproducible cell type-specific proteomics from individual mice. We also conducted bulk tissue proteomics on VM and striatum slices that were immediately frozen or subjected to APEX2 labeling procedures (*Figure 2—figure supplement 2a*). We found thousands of differentially expressed proteins when comparing VM vs. striatum slices, but no statistically significant differences when comparing acute vs. rested slices (*Figure 2—figure supplement 2b, c*). We also found that incubation of slices with 0.5 mM BP for 1 hr had no effect on spontaneous action potential frequency in mDA neurons (*Figure 2—figure supplement 2d, e*). These data demonstrate that acute slice preparation and labeling procedures do not compromise mDA neuronal function or significantly distort brain tissue proteomes.

To identify APEX2-dependent proteins captured by streptavidin pulldown, we normalized protein abundances to total protein intensity within each sample (see Materials and methods) and directly compared APEX2⁺ to APEX2⁻ control samples (*Figure 2e*). The number of proteins enriched in APEX2⁺ samples scaled with the fraction of biotinylation derived from each region (see *Figure 2b–d*), with 1449 proteins for VM, 702 proteins for MFB, 1840 proteins for striatal samples (FDR <0.05, Welch's unequal variance *t*-test with Benjamini–Hochberg correction). We emphasize that the normalized

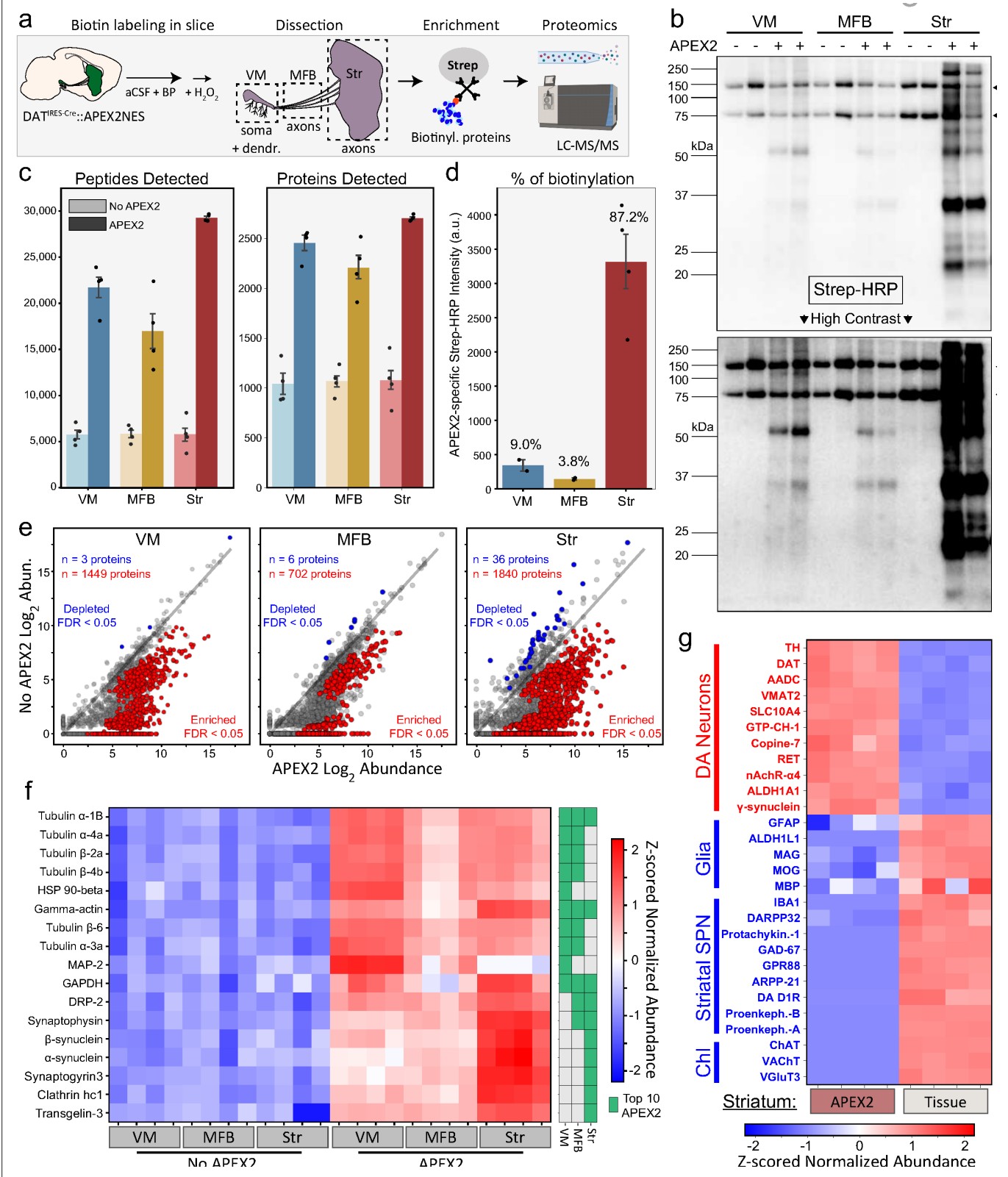

**Figure 2.** APEX2 proximity labeling proteomics in midbrain dopaminergic (mDA) neurons. (**a**) Schematic depicting APEX2 proximity labeling proteomics in mDA neurons. Slices are labeled, rapidly quenched, dissected, and flash frozen. Frozen tissues are lysed and precipitated to remove free biotin, after which resolubilized tissue proteins are subjected to streptavidin bead purification to enrich biotinylated proteins. On-bead digestion produces peptides which are quantified by liquid chromatography with tandem mass spectrometry (LC–MS/MS). (**b**) Streptavidin–HRP western blotting of streptavidin

*Figure 2 continued on next page*

*Figure 2 continued*

pulldowns from tissue dissections of the indicated regions (duplicate lanes are biological replicates). Each lane contains proteins eluted from 5% of the streptavidin beads for each biological replicate (single mouse/region). Arrows on the right indicate prominent bands in APEX2$^+$ and APEX2$^-$ samples of all regions, which represent endogenous biotinylated carboxylase proteins at ~75 and ~150 kDa. The majority of APEX2-specific biotinylation is found in the striatum, but specific labeling is present in both VM and MFB at high contrast (*lower*). (**c**) Mean ± standard error of the mean (SEM) of peptides and proteins detected per biological replicate of APEX2$^-$ or APEX2$^+$ streptavidin pulldowns of indicated regions (*n* = 4 each). See *Figure 2—source data 2* for raw label-free quantification intensity values of peptides and proteins for all samples used in this study. (**d**) Quantification of streptavidin–HRP reactivity in APEX2$^+$ streptavidin pulldowns, related to panel (**b**). After subtraction of endogenously biotinylated protein signal within each lane, the APEX2-specific streptavidin–HRP intensity for each APEX2$^+$ sample was determined by subtracting the average APEX2$^-$ lane intensity for the same region. Mean ± standard error of the mean (SEM) normalized streptavidin–HRP intensity is plotted for each region (*n* = 2 for VM, *n* = 2 for MFB, *n* = 4 for striatum). The percentage of APEX2-specific biotinylation found in each region are denoted above the bars. (**e**) Log–log abundance plots of APEX2$^-$ vs. APEX2$^+$ streptavidin pulldown samples for the indicated regions. Axes represent the average log$_2$(total intensity normalized abundance + 1) of *n* = 4 biological replicates for each sample type. Proteins significantly enriched or depleted from APEX2$^+$ streptavidin pulldown samples are colored in red or blue, respectively. False discovery rate (FDR) represents *q* values from Benjamini–Hochberg procedure on Welch's (unequal variance) *t*-test. See *Figure 2—source data 4* for complete results of APEX2$^+$ vs. APEX2$^-$ comparisons. (**f**) Heatmap of *Z*-scores for protein abundances for the union of the top 10 most abundant proteins enriched in APEX2$^+$ vs. APEX2$^-$ differential expression analysis from panel (**e**). Each column represents a biological replicate (*n* = 4) of APEX2$^-$ and APEX2$^+$ streptavidin pulldown samples in the indicated regions. The green color bar on the right indicates whether a given protein was in the top 10 of each region. (**g**) Heatmap of *Z*-scores for protein abundances for markers of mDA neurons, glia, striatal spiny projection neurons (SPNs), and cholinergic interneurons (ChI). Each column represents a biological replicate (*n* = 4) of bulk striatal tissue or APEX2$^+$ streptavidin pulldown samples. See *Figure 2—source data 5* for complete results of bulk tissue vs. APEX2$^+$ comparisons. *Abbreviations*: aCSF, artificial cerebrospinal fluid; BP, biotin phenol; H$_2$O$_2$, hydrogen peroxide; HRP, horseradish peroxidase; MFB, medial forebrain bundle; Str, striatum; VM, ventral midbrain. See *Figure 2—source data 1* for list of protein abbreviations in (**f, g**).

The online version of this article includes the following source data and figure supplement(s) for figure 2:

**Source data 1.** Protein abbreviations used in *Figure 2f, g*.

**Source data 2.** Raw label-free quantification intensity values of peptides and proteins for all samples used in this study.

**Source data 3.** Welch's *t*-test with BH false discovery rate (FDR) correction, related to *Figure 2—figure supplement 2b, c*.

**Source data 4.** Welch's *t*-test with BH false discovery rate (FDR) correction, related to *Figure 2e*.

**Source data 5.** Welch's *t*-test with BH false discovery rate (FDR) correction, related to *Figure 2g*.

**Source data 6.** Western blots related to *Figure 2b*.

**Figure supplement 1.** Proteomic depth and reproducibility.

**Figure supplement 2.** APEX2 slice labeling conditions do not distort bulk tissue proteomes.

proteomics data report on the *relative abundance* of proteins (relative to all protein captured by streptavidin pulldown in each sample), while the *absolute abundance* determined by western blot (*Figure 2b*) shows that most of the mDA neuronal protein mass is axonal. Although proteins such as endogenously biotinylated carboxylases (e.g., pyruvate carboxylase, propionyl-CoA carboxylase) and other nonspecific binders (e.g., myelin basic protein, proteolipid protein 1) were abundant in all samples, they were not enriched in APEX2$^+$ samples. The most abundant APEX2-specific proteins within each region displayed considerable overlap across regions (*Figure 2f*), with 17 common proteins derived from the top 10 proteins in each region (VM, MFB, and striatum). Although cytoskeletal proteins such as actin and tubulin subunits were highly abundant in APEX2$^+$ samples from all three regions, the most abundant APEX2-specific proteins within each region also included proteins enriched in specific subcellular compartments. For example, the somatodendritic protein MAP-2 (microtubule-associated protein 2) was in the top 10 only for VM samples (*Figure 2f*), while proteins involved in synaptic vesicle fusion and endocytosis were in the top 10 only for MFB or striatum samples (e.g., synaptophysin, synaptogyrin-3, and α-synuclein). Compared to bulk striatal tissue, APEX2$^+$ striatal samples show significant enrichment of dopaminergic proteins such as TH, dopamine transporter (DAT), aromatic-L-acid (dopa) decarboxylase (AADC), and vesicular monoamine transporter 2 (VMAT2) (*Figure 2g*). Meanwhile, proteins specific to astrocytes, microglia, oligodendrocytes, striatal spiny projection neurons (SPNs), and cholinergic interneurons (ChI) were either not detected or were significantly depleted from striatal APEX2$^+$ samples (*Figure 2g*). Thus, MS-based quantification of APEX2-enriched proteins enables proteomic profiling of the dopaminergic neuronal compartments contained within each brain region.

## Somatodendritic vs. axonal enrichment of proteins involved in diverse cellular functions

To further establish the compartment specificity of APEX2-specific proteins across regions, we first examined the abundance of the microtubule-binding proteins MAP-2 and tau, which are known to be enriched in somatodendritic and axonal compartments, respectively. Consistent with previous work (*Okabe and Hirokawa, 1989*; *Papasozomenos et al., 1985*), we found that MAP-2 was highly abundant in the VM and steadily decreased in the MFB and striatum, while tau was most abundant in striatal samples (*Figure 3a*). It should be noted that although we refer to VM samples as somatodendritic, these samples will necessarily include axons exiting the midbrain as well as somatically synthesized axonal proteins. Accordingly, the difference in abundance between striatal and VM samples was dramatically greater for MAP-2 compared to tau (*Figure 3a*). Thus, for our downstream analysis of somatodendritic and axonal compartments, we used the MFB samples only for filtering and focused primarily on VM and striatum samples.

To filter the APEX2$^+$ VM and striatum proteomics data, we took advantage of publicly available scRNA-seq data from the mouse VM and striatum (*Saunders et al., 2018*). After identification of high-confidence mDA neuron profiles, we used the mRNA expression distribution from scRNA-seq to establish a conservative lower bound for considering a gene as expressed in mDA neurons (*Figure 3— figure supplement 1a*, see Materials and methods). The vast majority of proteins in the APEX2 data were encoded by genes expressed in mDA neurons, with >93.0% of all detected proteins and >97.4% of APEX2-enriched proteins encoded by genes above our scRNA-seq threshold (*Figure 3—figure supplement 1b, c*). We removed proteins that were not significantly enriched in APEX2$^+$ > APEX2$^-$ samples, proteins encoded by genes below our mDA neuron scRNA-seq threshold, and proteins that did not show evidence of APEX2-specificity in at least two regions (*Figure 3—figure supplement 1d*, see Materials and methods for complete description). For comparison between VM and striatal samples, we retained the union of proteins passing filters in VM and striatal samples. Out of 1733 total proteins, 200 proteins passed filtering only in VM samples, 334 only in striatal samples, and 1199 both (*Figure 3b*). Hierarchical clustering of these 1199 overlapping proteins clearly segregated VM APEX2$^+$ and Str APEX2$^+$ samples from each other and from all APEX2$^-$ samples (*Figure 3—figure supplement 2*), and direct comparison between APEX2$^+$ samples revealed 173 and 374 proteins with greater relative abundance in VM or striatal samples, respectively (*Figure 3c*). Manual examination of VM- and Str-enriched protein clusters revealed striatal enrichment of synaptic vesicle proteins (e.g., Synaptotagmin-1 [Syt-1], Synaptophysin, SNAP25, VMAT2) and VM enrichment of postsynaptic scaffolding proteins (e.g., Homer-2, Shanks1-3) (*Figure 3—figure supplement 2*).

We first sought to confirm that cytoplasmic APEX2 labeling does not enrich proteins within membrane-enclosed structures. Among the 1733 total proteins passing filter in either VM or striatum, we found significant enrichment of gene ontology (GO) terms related only to nuclear and mitochondrial outer membranes (*Figure 3—figure supplement 3a, b*), but not nuclear inner membrane, nuclear lamina, nuclear matrix, mitochondrial inner membrane, or mitochondrial matrix. These results confirm the membrane impermeability of BP radicals and the integrity of organellar membranes in the acute brain slices.

To assess the relative enrichment of functionally related proteins within the somatodendritic and axonal compartments of mDA neurons, we conducted GO analysis of proteins that either passed filtering in only one region or were significantly enriched in that region in the APEX2$^+$ vs. APEX2$^-$ comparison (*Figure 3*, 373 proteins for VM, 708 for striatum). First, we analyzed subcellular localization GO terms curated by the COMPARTMENTS resource (*Binder et al., 2014*). We found that the top GO terms over-represented among VM-enriched proteins included 'postsynapse', 'somatodendritic compartment', 'dendrite', and 'postsynaptic density' (*Figure 3—figure supplement 3c*), while those over-represented among striatum-enriched proteins included 'presynapse', 'axon', 'synaptic vesicle', and 'axon terminus' (*Figure 3—figure supplement 3d*). Given the clear segregation of axonal/ presynaptic and dendritic/postsynaptic terms from the COMPARTMENTS resource in striatum- and VM-enriched proteins, respectively, we analyzed the VM- and Str-enriched proteins using the GO Consortium resource (*The Gene Ontology Consortium, 2021*). Similar to the COMPARTMENTS analysis, we found that GO terms related to postsynaptic function were over-represented in VM-enriched proteins, with terms such as 'dendrite', 'glutamate receptor binding', and 'GABA-A receptor complex'. Canonical proteins associated with these terms included subunits and scaffolding proteins

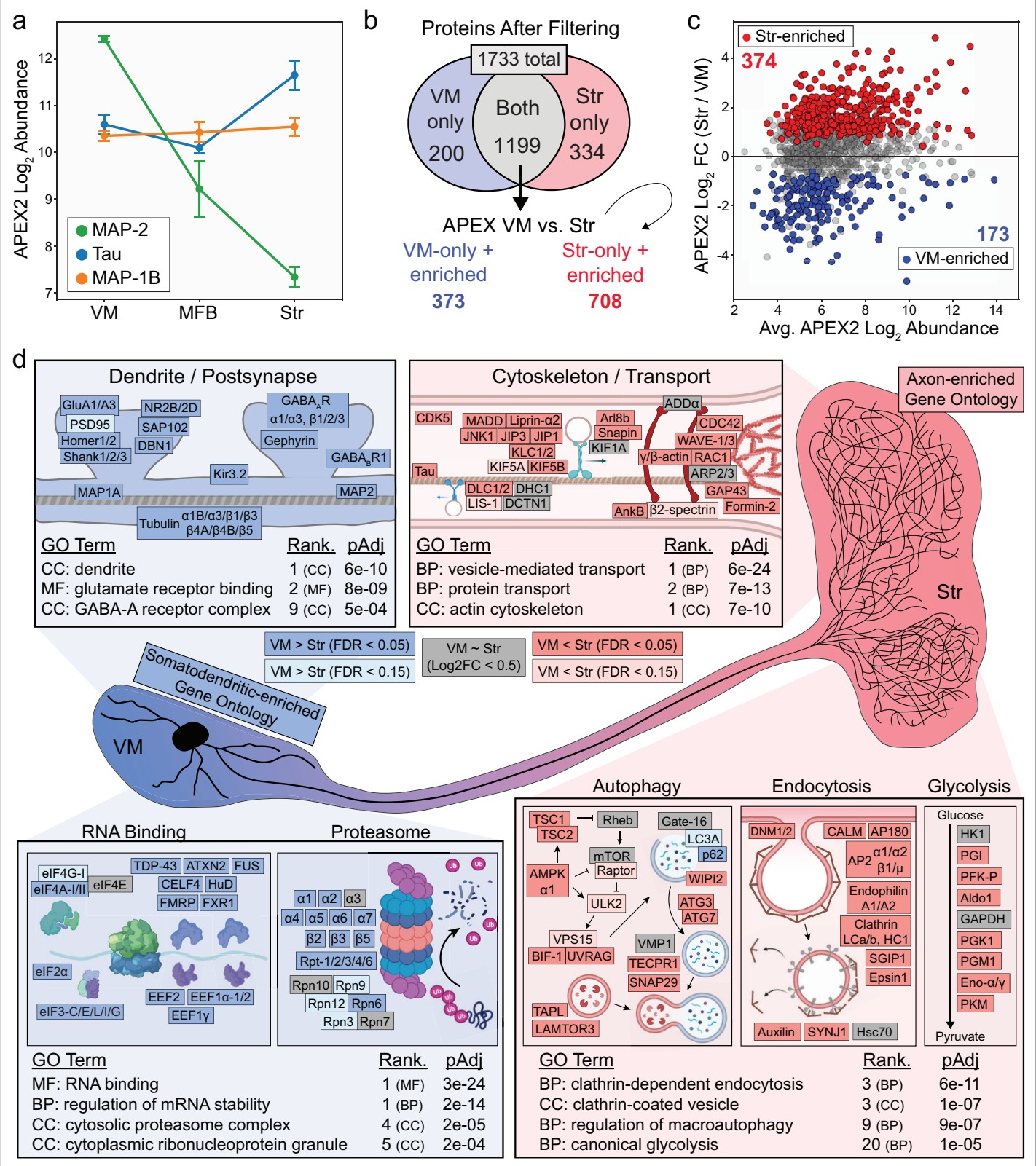

**Figure 3.** Axonal and somatodendritic proteomics reveals gene ontologies enriched in subcellular compartments. (**a**) APEX2 proteomics data for microtubule-associated proteins MAP-2, Tau, and MAP-1B. Mean ± standard error of the mean (SEM) of the protein abundances, as $\log_2$(total intensity normalized abundance + 1), are shown for $n$ = 4 biological replicates of APEX2+ streptavidin pulldown samples in the indicated regions. (**b**) Schematic depicting proteins remaining after filtering. For complete filtering workflow, see Materials and methods and *Figure 3—figure supplement 1*. Proteins

*Figure 3 continued on next page*

*Figure 3 continued*

present in both VM and Str after filtering were further compared by differential expression—see panel (**c**). Proteins present only in VM or Str, plus those enriched in VM vs. Str differential expression analysis, were used for subsequent gene ontology (GO) analysis. (**c**) Differential expression comparison of VM vs. Str APEX2⁺ streptavidin pulldown samples. Proteins colored red or blue had a false discovery rate (FDR) <0.05 after Benjamini–Hochberg corrected p values from Welch's (unequal variance) *t*-test. See *Figure 3—source data 2* for complete results and summary of proteins before and after filtering. (**d**) GO analysis (*Enrichr*) of VM- and Str-enriched proteins (373 and 708, respectively, see panel b). Selected GO terms are listed along with adjusted p values and adjusted p value rank for each GO term category (Cellular Component, Molecular Function, Biological Process). Canonical and representative proteins from the ontologies are shown. Every protein depicted is present in the filtered proteomics data of VM, Str, or both. Colors indicate significant (dark red/blue) enrichment, near-significant enrichment (light red/blue), or similar levels between VM and Str (gray). Slashes indicate separate proteins (e.g., DLC1/2 represents both DLC1 and DLC2). See *Figure 3—source data 3* for complete GO summary. *Abbreviations*: MAP-2, microtubule-associated protein 2; Tau, microtubule-associated protein tau; MAP-1B, microtubule-associated protein 1B; MFB, medial forebrain bundle; Str, striatum; VM, ventral midbrain. See *Figure 3—source data 1* for list of protein abbreviations in (**d**).

The online version of this article includes the following source data and figure supplement(s) for figure 3:

**Source data 1.** Protein abbreviations used in *Figure 3d*.

**Source data 2.** Summary of all APEX2 comparisons and single-cell RNA-sequencing (scRNA-seq) data for all proteins and proteins retained after filtering, related to *Figure 3—figure supplement 1*.

**Source data 3.** Enrichr gene ontology (GO) analysis related of ventral midbrain (VM)-enriched, and striatum-enriched proteins, related to *Figure 3d* and *Figure 3—figure supplement 3c, d*.

**Figure supplement 1.** Filtering of APEX2 proteomics data using single-cell RNA-sequencing (scRNA-seq) data and cross-regional comparisons.

**Figure supplement 2.** Clustered heatmap of protein abundances for union of filtered ventral midbrain (VM) and Str APEX2 proteins.

**Figure supplement 3.** Subcellular compartment gene ontology (GO) analysis of filtered ventral midbrain (VM) and Str APEX2 proteins.

for AMPA, NMDA, GABA-A, and GABA-B receptors (*Figure 3d*). GO terms such as 'RNA binding' and 'regulation of mRNA stability' were also over-represented in VM-enriched proteins (*Figure 3d*), consistent with the somatodendritic compartment being the major site of protein synthesis in neurons. Despite a clear role for the ubiquitin proteasome system in axons (*Korhonen and Lindholm, 2004*), we found over-representation of GO terms such as 'cytosolic proteasome complex' in VM-enriched proteins, with nearly all 20S subunits and most 19S subunits showing higher relative abundance in VM APEX2⁺ samples (*Figure 3d*).

The enrichment of protein synthesis and degradation machinery within the somatodendritic compartment is consistent with our recent work on local translation in mDA neurons (*Hobson et al., 2021*) and underscores the importance of cytoskeletal transport systems in axonal protein homeostasis (*Maday et al., 2014*; *Roy, 2014*). Accordingly, GO terms such as 'vesicle-mediated transport' and 'protein transport' were over-represented in striatum-enriched proteins. Striatum-enriched proteins in these ontologies included kinesin and dynein subunits, cargo adaptor proteins, and upstream kinases involved in transport regulation (*Figure 3d*). In addition to microtubule-based transport proteins, proteins related to the axonal actin cytoskeleton were also over-represented in striatum APEX2⁺ samples. These included actin subunits themselves, GTPases and other regulators of actin nucleation, and proteins that link actin rings in the distal axon (*Leterrier et al., 2017*; *Xu et al., 2013*; *Figure 3d*). We also found that proteins involved in clathrin-dependent endocytosis were uniformly enriched in striatum APEX2⁺ samples, consistent with high rates of synaptic vesicle recycling in mDA neurons related to their tonic activity. Given the intense energetic demands placed on mDA axons (*Pissadaki and Bolam, 2013*), we were intrigued to find an over-representation of glycolytic enzymes in Str-enriched proteins: seven out of nine glycolytic enzymes show higher relative abundance in striatum compared to VM (*Figure 3d*). These results suggest that glycolysis may be especially important in dopaminergic axonal metabolism, consistent with glycolysis supporting axonal transport and presynaptic function in other neurons (*Ashrafi et al., 2017*; *Hinckelmann et al., 2016*; *Jang et al., 2016*; *Zala et al., 2013*). We note that striatum APEX2⁺ samples also showed extensive enrichment of GO terms related to presynaptic function and synaptic vesicle proteins, as detailed below (see *Figure 3—source data 3* for a complete GO analysis summary).

While anterograde transport is critical for delivery of new proteins to distal axons, it is likely that both local autophagy and retrograde transport contribute to protein clearance in mDA axons. Many autophagosomes formed in distal axons are transported back to the soma prior to fusion with lysosomes (*Maday et al., 2012*), although other findings implicate local autophagic degradation of damaged mitochondria in axons (*Ashrafi et al., 2014*). Indeed, we previously showed that macroautophagy

regulates presynaptic structure and function in mDA axons (*Hernandez et al., 2012*). Consistent with these results, we found that GO terms related to autophagy were over-represented in Str-enriched proteins (*Figure 3d*). Autophagy-related proteins enriched in striatum APEX2⁺ samples included kinases involved in upstream regulation of autophagy, membrane proteins involved in autophagosome maturation, and membrane proteins involved in lysosomal membrane fusion. Among the autophagy-related proteins present in striatal axons, vacuole membrane protein 1 (VMP1) was recently shown to be critical for survival and axonal integrity in mDA neurons (*Wang et al., 2021*). Our results suggest that VMP1 may regulate autophagy in the axons as well as soma of mDA neurons, highlighting the ability of APEX2 proteomics to elucidate the distribution of proteins with unknown localization. Collectively, these results establish a foundation of protein localization that underlies diverse cellular functions within the somatodendritic and axonal compartments of mDA neurons. While our GO analysis is broadly consistent with established features of neuronal polarization (e.g., somatodendritic enrichment of terms such as 'postsynapse' and 'RNA binding'), findings from this discovery approach will require follow-up studies to elucidate compartment-specific function of the identified proteins.

## Localization of K⁺ channels Kv4.3 and GIRK2 in dopaminergic, but not cortical axons

To validate our axonal vs. somatodendritic comparisons using orthogonal approaches, we first used western blotting to compare the axonal or somatodendritic enrichment of TH, βIII-tubulin, and synaptophysin (*Figure 4a*). We found a strong correlation between the Str-to-VM abundance ratios computed from proteomics and western blot analysis ($r = 0.998$, *Figure 4b*). We next compared our axonal proteome to a recent dataset generated from cultured cortical axons (*Chuang et al., 2018*). Despite significant differences in the species (mouse vs. rat), neuronal population (mDA vs. cortical neurons), state of maturity (adult mice vs. embryonic neurons in vitro), and axonal protein isolation methods (APEX2 labeling vs. microfluidic isolation), we found 978 proteins detected in both datasets and a significant correlation between their abundances ($r = 0.56$, $p < 3e{-}80$, *Figure 4—figure supplement 1a, b*). Although this overlap is likely an underestimate due to the differences noted above, these data suggest that at least half of our axonal proteome is conserved in cortical axons.

We conducted GO analysis of proteins unique to each dataset. In addition to mDA neuron-enriched proteins (i.e., TH, DAT, VMAT2), proteins related to cytoskeleton, protein transport, and SV release/recycling were over-represented for proteins unique to the mDA data (*Figure 4—figure supplement 1c*). In contrast, GO analysis of proteins unique to the cortical axon data revealed overrepresentation of proteins related to gene expression, RNA binding, and translation, including 41 large ribosomal subunit and 27 small ribosomal subunit proteins (*Figure 4—figure supplement 1d*). Although we have recently demonstrated a striking absence of axonal translation in mDA neurons (*Hobson et al., 2021*), these findings also highlight technical differences between the two datasets, since many ribosomal proteins would not have surface residues accessible to cytoplasmic APEX2 labeling. Indeed, we also observed a significant overrepresentation of proteins associated with 'intracellular organelle lumen' and 'mitochondrial matrix' in the cortical axon data (*Figure 4—figure supplement 1d*). Nonetheless, the presence of histone proteins in cortical axons (*Chuang et al., 2018*) appears not to be conserved in mature mDA axons: thus, some features of these axonal proteomes are likely cell type specific.

Among proteins unique to our striatal APEX2 data, we found that the A-type K⁺ channel Kv4.3 (*Kcnd3*) and the G-protein-regulated inward-rectifier potassium channel 2 (GIRK2;*Kcnj6*) were present in the VM, MFB, and striatum (*Figure 4c*). Both of these potassium channels are known to be important for mDA neuronal physiology: Kv4.3 channels mediate somatodendritic A-type K⁺ currents ($I_A$) that regulate pacemaker frequency (*Haddjeri-Hopkins et al., 2021*; *Liss et al., 2001*; *Serôdio and Rudy, 1998*), while GIRK2 mediates somatodendritic hyperpolarization downstream of D2 auto-receptors (*Beckstead et al., 2004*; *Ford, 2014*). However, the APEX2 labeling of Kv4.3 and GIRK2 from the MFB and striatum was surprising, given that these channels are densely localized within the somatodendritic compartment (*Reyes et al., 2012*; *Rhodes et al., 2004*) and assumed to mediate postsynaptic rather than presynaptic functions (*Lüscher et al., 1997*; *Martel et al., 2011*). We therefore used confocal immunofluorescence to confirm the axonal localization of GIRK2 and Kv4.3 in sagittal sections. As expected, we observed strong labeling of somatodendritic membranes for both proteins in mDA neurons within the VM (*Figure 4d, e*, *left*). Labeling in the MFB was less intense, but clear colocalization along the course of TH⁺ axons was apparent at high magnification (*Figure 4d, e*,

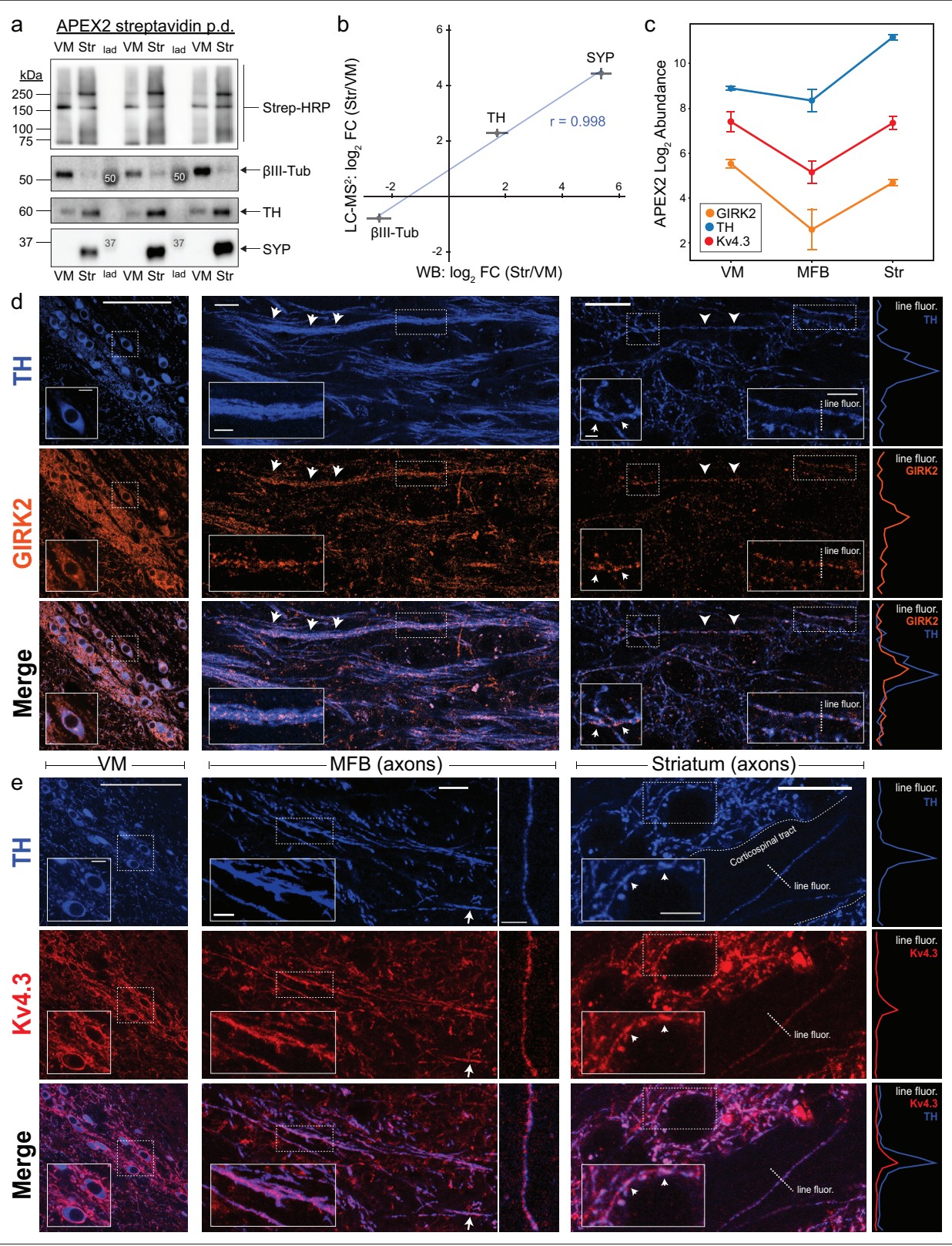

**Figure 4.** Antibody validation and axonal localization of potassium channels Kv4.3 and GIRK2. (**a**) Western blotting of streptavidin pulldowns from APEX2[+] samples, with proteins eluted from an equal fraction of streptavidin beads (~2.5% of total captured protein in each region) in each lane. Ventral midbrain (VM) and Str pairs from the same mouse are separated by molecular weight ladder (lad). Molecular weight markers detected by chemiluminescence are indicated with white font in the images. (**b**) Quantification of proteomics (LC–MS[2]) and western blot (WB) data for TH, SYP

*Figure 4 continued on next page*

Figure 4 continued

(synaptophysin), and βIII-tubulin (*Tubb3*). Mean ± standard error of the mean (SEM) of the $\log_2$ FC (Str/VM) from $n = 4$ biological replicates. Pearson's correlation coefficient *r* as indicated (p < 0.05). (**c**) APEX2 proteomics data for GIRK2 (Kir3.2/*Kcnj6*), TH, and Kv4.3 (*Kcnd3*). Mean ± SEM of the protein abundances, as $\log_2$(total intensity normalized abundance + 1), are shown for $n = 4$ biological replicates of APEX2$^+$ streptavidin pulldown samples in the indicated regions. Immunohistochemistry for TH and potassium channels GIRK2 (**d**) or Kv4.3 (**e**) in sagittal sections. *Left*: VM, soma, and dendrites. *Middle*: medial forebrain bundle (MFB), axons. *Right*: striatum, axons. Insets in each image are indicated with dashed white rectangles. Arrows indicate prominent sites of colocalization. Traces on the far right are fluorescence profiles for the indicated dashed lines ('line fluor'). (**d**) Scale bars: (*Left*) main: 100 μm, inset: 10 μm. (*Middle*) main: 15 μm, inset: 5 μm. (*Right*) main: 15 μm, left inset: 2 μm, right inset: 5 μm. (**e**) Scale bars: (*Left*) main: 100 μm, inset: 10 μm. (*Middle*) main left: 15 μm, inset: 10 μm, main right: 10 μm. (*Right*) main: 15 μm, inset: 5 μm.

The online version of this article includes the following source data and figure supplement(s) for figure 4:

**Source data 1.** Western blots related to *Figure 4a*.

**Source data 2.** Enrichr GO analysis related to *Figure 4—figure supplement 1*.

**Figure supplement 1.** Comparison of striatal APEX2 midbrain dopaminergic (mDA) axon proteome and cortical axon proteome.

**Figure supplement 2.** Somatodendritic enrichment of Homer2 confirmed via immunohistochemistry.

*middle*). Similarly, scattered labeling of the striatal neuropil was colocalized with TH$^+$ axonal fibers and varicosities (*Figure 4d, e*, *right*). Thus, our APEX2 data accurately predict that both Kv4.3 and GIRK2 are present in striatal mDA axons, but are not enriched relative to the somatodendritic compartment.

In some instances, we observed a few mDA axons crossing myelinated corticospinal tracts that course through the striatum (*Figure 4e*, *right*). Kv4.3 immunoreactivity within these tracts was clearly restricted to TH$^+$ axonal fibers, indicating that Kv4.3 is present in mDA axons, but not corticospinal axons. These results are consistent with the absence of Kv4.3 in the cortical axon proteomic data (*Chuang et al., 2018*) and ultrastructural studies of Kv4.3 localization in the mouse visual cortex (*Burkhalter et al., 2006*). Future studies are required to determine the functional role of axonal GIRK2 and Kv4.3 in mDA neurons, but the axonal localization of these K$^+$ channels should be considered when interpreting knockout studies (*Costa et al., 2021*; *McCall et al., 2017*). More broadly, these data suggest that some, but not all, somatodendritic proteins are also in mDA axons. Indeed, consistent with the APEX2 data, we found that immunoreactivity for Homer2, an mDA neuron-enriched postsynaptic protein, is prominently localized within dopaminergic dendrites in the SNr but not within MFB/Str axons (*Figure 4—figure supplement 2*).

**The dopaminergic presynaptic proteome via APEX2 labeling in striatal slices and synaptosomes**
mDA neuron-specific APEX2 labeling in striatal slices provides a convenient means to isolate proteins from dopaminergic axons, but acute slice preparation may not be practical for all laboratories. To further study the presynaptic proteome of dopaminergic axons and provide an alternative to the slice labeling procedure, we conducted APEX2 labeling in synaptosomes prepared from the striatum of DAT-IRES-Cre mice expressing APEX2NES (*Figure 5a*). Synaptosomes are resealed nerve terminals formed by liquid shearing forces during homogenization of brain tissue in isotonic sucrose buffer (*Gray and Whittaker, 1962*; *Whittaker, 1993*). After a brief, low-speed centrifugation to remove heavy cellular debris and nuclei, the majority of synaptosomes can be rapidly recovered in the pellet from moderate-speed centrifugation known as the P2 fraction. Although the P2 fraction also contains myelin and free mitochondria, we reasoned that these contaminants would not interfere with APEX2 labeling within resealed dopaminergic nerve terminals. After isolation and washing of the P2 fraction, reagents were added directly to synaptosomes under typical in vitro APEX2 labeling conditions (30-min incubation with 0.5 mM BP followed by 60 s of 1 mM $H_2O_2$). Similar to the slice procedure, streptavidin–HRP western blotting of striatal P2 lysates showed APEX2-dependent protein biotinylation across a wide range of molecular weights (*Figure 5b*). Staining of labeled P2 samples with fluorescent streptavidin revealed deposition of biotin within TH$^+$/V5-APEX2$^+$ particles approximately ~1 μm in diameter, consistent with APEX2 labeling within dopaminergic synaptosomes (*Figure 5b*). We analyzed APEX2-enriched proteins from striatal synaptosomes by MS. As before, we controlled for nonspecific binding by conducting all labeling and protein purification procedures on striatal synaptosomes from mice with and without APEX2 expression in mDA neurons.

Coverage of both peptides and proteins in APEX2$^+$ synaptosome samples was remarkably similar to the APEX2$^+$ striatal slice samples (*Figure 5c*), although nonspecific binding was slightly higher in APEX2$^-$ synaptosome samples. Principal component analysis revealed tight cosegregation of APEX2$^+$ striatal slice and synaptosome samples apart from all other samples, suggesting these APEX2$^+$

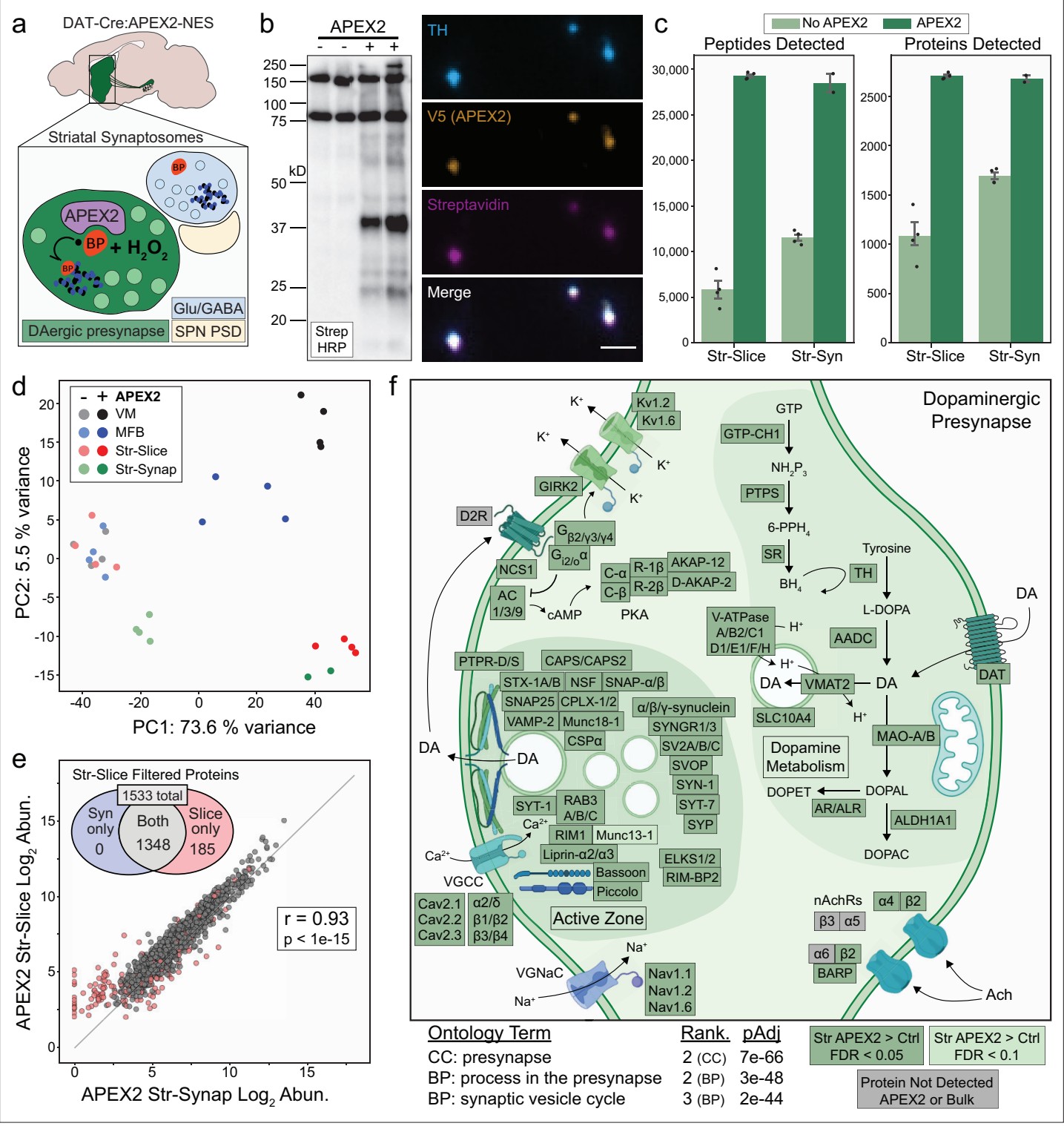

**Figure 5.** APEX2 labeling in synaptosomes and dopaminergic presynaptic proteome. (**a**) Schematic depicting APEX2 labeling in dopaminergic striatal synaptosomes. A crude synaptosome fraction (P2) is rapidly prepared from the striatum, washed, incubated with biotin phenol (0.5 mM for 30 min), labeled with 1 mM hydrogen peroxide ($H_2O_2$) for 60 s, quenched with antioxidants and sodium azide, centrifuged, and flash frozen as a pellet for downstream streptavidin pulldown and proteomics (see Materials and methods). (**b**) *Left*: streptavidin–horseradish peroxidase (HRP) western blot of proteins captured by streptavidin pulldown from striatal synaptosomes of DAT-Cre mice with or without expression of APEX2-NES. *Right*: immunostaining of synaptosomes from DAT-Cre:APEX2+ striatum. Synaptosomes were bound to poly-lysine coated coverslips during biotin phenol incubation, washed, and fixed after $H_2O_2$ treatment. Scale bar: 2 µm. (**c**) Mean ± standard error of the mean (SEM) of peptides and proteins detected per

*Figure 5 continued on next page*

*Figure 5 continued*

biological replicate of APEX2⁻ or APEX2⁺ streptavidin pulldowns of indicated regions. Str-slice data are the same as in same as in **Figure 2c** (n = 4 each for APEX2⁻/APEX2⁺). n = 4 for APEX2⁻ Str-Syn, n = 2 for APEX2⁺ Str-Syn. (**d**) Principal components analysis of all APEX2⁻ and APEX2⁺ biological replicates for the 1733 proteins present either in VM or Str after filtering (see **Figure 3b**). PC1 captures 73.6% of the variance and is dominated by APEX2⁺ vs. APEX2⁻ samples, while PC2 captures 5% of the variance and stratifies regional APEX2⁺ samples. APEX2⁺ Str-Syn samples are highly similar to APEX2⁺ Str-Slice samples. (**e**) Log–log abundance plots of APEX2⁺ streptavidin pulldown samples for the indicated regions. Axes represent the average $\log_2$(total intensity normalized abundance + 1) for each sample type (n = 2 and n = 4 biological replicates for Str-Synaptosomes and Str-Slice, respectively). Out of 1533 Str-Slice APEX2⁺ filtered proteins (see **Figure 3—figure supplement 1d**), 1348 show concordant enrichment in Str-Syn APEX2⁺ vs. APEX2⁻ comparisons ($\log_2$ FC >0 and false discovery rate [FDR] <0.15). These 1348 proteins were retained for gene ontology (GO) analysis (panel f). Accordingly, Str-Syn and Str-Slice APEX2⁺ samples show robust Pearson's correlation (r = 0.93, p value <1e−15). (**f**) GO analysis (*SynGO*) of Str-Slice- and Str-Syn-enriched proteins 1348 proteins, see panel (**e**). Selected GO terms are listed along with adjusted p values and adjusted p value rank for each GO term category (Cellular Component, Biological Process). Canonical and representative proteins from the ontologies are shown. Proteins depicted are either present in the filtered Str-Slice proteomics data (FDR <0.05, APEX2 vs. Control, dark green), nearly missed the Str-Slice significance thresholds (FDR <0.1, APEX2 vs. Control, light green), or were not detected in any APEX2 or bulk striatal tissue samples (gray). See **Figure 5—source data 2** for complete GO summary. *Abbreviations*: SPN PSD, spiny projection neuron postsynaptic density; TH, tyrosine hydroxylase; MFB, medial forebrain bundle; Str, striatum; VM, ventral midbrain. See **Figure 5—source data 1** for complete list of protein and metabolite abbreviations in (**f**).

The online version of this article includes the following source data and figure supplement(s) for figure 5:

**Source data 1.** Protein and metabolite abbreviations used in **Figure 5f**.

**Source data 2.** SynGO analysis related to **Figure 5f** and **Figure 5—figure supplement 1**.

**Source data 3.** Protein abbreviations used in **Figure 5—figure supplement 1**.

**Source data 4.** Western blots related to **Figure 5b**.

**Figure supplement 1.** Synaptic gene ontology analysis proteins of striatal slice and synaptosomal APEX2 data.

**Figure supplement 2.** APEX2 enrichment of specific protein groups across midbrain dopaminergic (mDA) neuronal regions.

proteomes are highly similar (**Figure 5d**). Indeed, the correlation between APEX2⁺ striatal slice and synaptosome samples was comparable to that of biological replicates within each sample type (Pearson's r = 0.93, p < 1e−15) (**Figure 5e**). Out of 1533 proteins that passed filtering in striatal slices, 1348 of these showed enrichment in APEX2⁺ vs. APEX2⁻ synaptosomes (**Figure 5e**).

Proteins involved in DA metabolism were significantly enriched in both slice and synaptosome APEX2⁺ samples, including canonical DA synthesis, release, and reuptake proteins (TH, AADC, VMAT2, and DAT) as well as enzymes involved in tetrahydrobiopterin synthesis (GTP cyclohydrolase I, 6-pyruvoyltetrahydropterin synthase, and sepiapterin reductase) and DA degradation (monoamine oxidase A/B, retinaldehyde dehydrogenase 1, aldose reductase, and aldehyde reductase) (**Figure 5f**). Thus, all of the protein machinery required for DA neurotransmission and metabolism is present within dopaminergic axonal boutons, including monoamine oxidases that metabolize DA and contribute to oxidative phosphorylation via presynaptic mitochondria (**Graves et al., 2020**). We also found significant enrichment of proteins that contribute to DA vesicular uptake, including most subunits of the vesicular ATPase and SLC10A4, an orphan transporter found on monoaminergic synaptic vesicles that contribute to axonal DA homeostasis (**Larhammar et al., 2015**; **Figure 5f**).

Striatal DA release is powerfully modulated by nicotinic receptors and DA D2 receptors on dopaminergic axons (**Sulzer et al., 2016**). Consistent with pharmacological studies (**Exley and Cragg, 2008**), we found that β2 and α4 nicotinic receptor subunits were enriched by APEX2 in striatal slices and synaptosomes (**Figure 5f**). Despite a wealth of functional evidence supporting their presence on mDA axons, the DA D2 receptor as well as β3, α5, and α6 nicotinic receptor subunits were notably absent from our dataset. Since the vast majority of APEX2 labeling occurs on tyrosine residues (**Kim et al., 2018**; **Udeshi et al., 2017**), some membrane proteins may not be labeled if they lack cytoplasm-facing tyrosine residues accessible to BP radicals. However, the absence of these nicotinic receptor subunits and the DA D2 receptor is likely related to MS, since none of these membrane proteins were detected in any bulk striatal tissue sample (**Figure 5f**, gray boxes). Thus, the absence of a protein in our dataset does not necessarily indicate the absence in mDA axons, especially when other functionally related proteins are enriched. For example, we found significant APEX2 enrichment of β-anchoring and -regulatory protein (BARP), a protein recently shown to interact with and modulate α6/β2/β3 nicotinic receptors on dopaminergic axons (**Gu et al., 2019**; **Figure 5f**). Similarly, we find significant APEX2 enrichment of D2 receptor-interacting proteins (e.g., G protein subunits and neuronal calcium sensor NCS-1) and downstream effectors (e.g., adenylyl cyclase and PKA subunits) known to function

within mDA neurons (*Dragicevic et al., 2014*). We also found significant enrichment of voltage gated potassium channels Kv1.2 and Kv1.6, consistent with previous work on Kv1 channels as downstream effectors of presynaptic D2 receptor function (*Martel et al., 2011*; *Figure 5f*).

We next focused on the dopaminergic presynapse, using the SynGO resource (*Koopmans et al., 2019*) to analyze the 1,348 proteins enriched in APEX2+ striatal slice and synaptosome samples. Ontology terms such as 'presynapse', 'synaptic vesicle', and 'presynaptic active zone' were all highly over-represented among APEX2-enriched proteins (*Figure 5f* and *Figure 5—figure supplement 1a*). Consistent with recent work on the molecular architecture of striatal DA release sites, APEX2 labeling of striatal and synaptosome samples significantly enriched active zone proteins RIM1, Bassoon, Liprin-α2 and -α3, Munc13-1, ELKS1/2, and RIM-BP2 (*Figure 5f*), although ELKS1/2 and RIM-BP2 are not essential for DA release (*Banerjee et al., 2020a*; *Liu et al., 2018*). Coverage of synaptic vesicle trafficking and fusion proteins was extensive, with significant enrichment of all major integral synaptic vesicle proteins (SV2A-C, CSPα, Synaptogyrin1–3, Synaptophysin, Synapsin-1, RAB3A-C, VAMP-2), target SNAREs (Syntaxin 1A/B, SNAP25), and functionally related proteins (NSF, SNAP-α/β, Munc18-1, Complexin1/2, and α/β/γ-synuclein). We also observed a significant enrichment of CAPS1 and CAPS2, proteins that prime synaptic vesicles in hippocampal neurons (*Jockusch et al., 2007*) and regulate catecholamine loading into large dense-core vesicles in adrenal chromaffin cells (*Speidel et al., 2005*) but whose function is to our knowledge unexplored in mDA neurons. Notably, we find significant enrichment of Synaptotagmins-1 and -7 (Syt-1/Syt-7) (*Figure 5f*). These data support the recent finding that Syt-1 is the major fast $Ca^{2+}$ sensor for synchronous DA release (*Banerjee et al., 2020b*). In addition to the established role of Syt-7 in somatodendritic DA release and the observation of its presence in DA axon terminals (*Delignat-Lavaud et al., 2021*; *Mendez et al., 2011*), our findings support Syt-7 as a candidate protein for asynchronous axonal DA release (*Banerjee et al., 2020b*). These data highlight candidate proteins for further investigation and provide a proteomic architecture of the dopaminergic presynapse that is highly consistent with recent functional studies (*Banerjee et al., 2020a*; *Banerjee et al., 2020b*; *Liu et al., 2018*).

Somewhat unexpectedly, we also found significant APEX2 enrichment of proteins with SynGO annotations for ontology terms related to postsynaptic function (*Figure 5—figure supplement 1a*). Many of these proteins were annotated for both pre- and postsynaptic function, including proteins such as DAT and VMAT2 (*Figure 5—figure supplement 1b*). Although these proteins are present within dopaminergic dendrites (*Nirenberg et al., 1996a*; *Nirenberg et al., 1996b*), they are clearly derived from dopaminergic axons in our striatal APEX2+ samples. Similarly, many proteins with postsynaptic annotations were functionally related to actin dynamics, PKA or MTOR signaling, or other functions not exclusively related to postsynaptic function. Other proteins bearing only postsynaptic SynGO annotations, such as the Netrin receptor DCC and metabotropic glutamate receptor mGluR1, have functional evidence supporting their localization on mDA axons (*Reynolds et al., 2018*; *Zhang and Sulzer, 2003*). Nonetheless, 63 proteins with clear roles in dendrites were enriched by APEX2 in both slice and synaptosome samples (*Figure 5—figure supplement 1b*). Two-thirds of these proteins were also present in cortical axons in vitro (*Chuang et al., 2018*), and they were significantly lower in abundance than proteins with presynaptic annotations in both datasets (*Figure 5—figure supplement 1c*). Despite significant biological and technical differences in these two studies, the correlation of protein abundance was similar for proteins with pre- or postsynaptic annotation (*Figure 5—figure supplement 1d*). Although we cannot rule out the possibility of low levels of post-synaptic contamination, the significant depletion of proteins highly expressed by glia and striatal neurons (*Figure 2e*) demonstrates the absence of extensive cross-membrane labeling. Furthermore, we observed no enrichment of DARPP-32 and Spinophilin (*Figure 5—figure supplement 2a*), proteins highly abundant within striatal dendritic spines (*Allen et al., 1997*; *Blom et al., 2013*; *Greengard et al., 1999*; *Smith et al., 1999*).

APEX2 enrichment in all regions can provide additional evidence of axonal localization for proteins often found in the postsynapse. For example, GABA receptors are typically present on dendrites, but recent electrochemical and electrophysiological studies have demonstrated the presence of both GABA-A and GABA-B receptors on mDA axons (*Kramer et al., 2020*; *Lopes et al., 2019*; *Schmitz et al., 2002*). We found that GABA-A receptor subunits and scaffolding protein gephyrin were most abundant in VM APEX2+ samples, but still higher in APEX2+ than APEX2− samples from both MFB and striatum (*Figure 5—figure supplement 2b*). Similar results were obtained for GABA-B receptor

subunits, the PDZ scaffold Mupp1 (*Balasubramanian et al., 2007*; *Figure 5—figure supplement 2c*). Although many comparisons did not pass the FDR correction in MFB comparisons, likely due to substantially lower proteomic depth in these samples, APEX2 enrichment in both MFB and striatal samples supports axonal transport and localization of these proteins.

Axonal APEX2 proteomics allowed us to identify proteins and protein complexes that are previously undescribed in striatal mDA axons. The cytosolic chaperonin T-complex protein 1-ring complex (TRiC), or chaperonin containing T-complex (CCT) is an oligomeric complex that promotes folding of newly synthesized polypeptides, suppresses aggregation of huntingtin in Huntington's disease, and can inhibit assembly of α-synuclein amyloid fibrils (*Lopez et al., 2015*; *Sot et al., 2017*; *Tam et al., 2009*). Recent work has implicated specific TRiC/CCT subunits in the regulation of axonal transport in cortical neurons (*Chen et al., 2018*; *Zhao et al., 2016*), but axonal localization of TRiC subunits is largely undescribed. We detected seven out of eight TRiC subunits in our APEX2 proteomics data, six of which showed strong evidence of axonal localization (*Figure 5—figure supplement 2d*). These results suggest that the recently described interaction between CCT5/CCTε, CDK5, and tau (*Chen et al., 2018*) may regulate retrograde transport in mDA neurons. Given that TRiC/CCT can regulate α-synuclein aggregation (*Sot et al., 2017*), future research on TRiC/CCT function in dopaminergic axons is warranted.

## Subcellular localization of proteins encoded by mDA neuron-enriched genes

Genetic analysis can identify mutations that cause familial PD or variants linked to sporadic PD risk, but not whether the relevant genes are expressed in mDA neurons. Although TRAP and scRNA-seq provide significant insight into mDA neuronal gene expression, these techniques do not address the subcellular localization of the encoded proteins. We leveraged our APEX2 proteomics data to interrogate proteins encoded by genes with high DA neuron specificity (*Figure 6a*). We reanalyzed published scRNA-seq data (*Saunders et al., 2018*; *Figure 6—figure supplement 1*), and identified 64 genes with >eightfold higher expression in mDA neurons compared to all other midbrain cells (see Materials and methods). Fifty-five proteins encoded by these genes were present in the filtered APEX2 proteomics data (*Figure 6—source data 1*).

Only a minority of proteins encoded by mDA neuron-enriched were preferentially localized to the somatodendritic compartment of mDA neurons (*Figure 6b*). In contrast, a majority of proteins encoded by mDA neuron-enriched genes (35 out of 55, *Figure 6b*) were preferentially localized to dopaminergic axons (FDR <0.05 for APEX2+ striatal vs. VM samples). mDA neuron-enriched genes encoding striatal APEX2-enriched proteins included many canonical synaptic vesicle and active zone proteins mentioned above (e.g., Bassoon, Syt-1, Complexin-1/2, SV2B/C, RAB3C, CAPS2), highlighting a striking proportion of mDA neuronal gene expression dedicated to presynaptic function (*Figure 6b*). Another group of APEX2+ striatum-enriched proteins were those involved in DA synthesis and transmission, including AADC, TH, DAT, and GTP cyclohydrolase I (*Figure 6b*).

The axonal enrichment of Syt-17 is also of particular interest, given that human SYT17 lies within a PD risk locus (*Nalls et al., 2019*) and *Syt17* mRNA is enriched in mDA neurons (*Figure 6b*). Syt-17 is an atypical synaptotagmin that does not bind calcium or participate in synaptic vesicle fusion (*Ruhl et al., 2019*) and has no established role in axons. Although hippocampal neurons from Syt-17 knockout mice display axonal growth defects, tagged Syt-17 is found in the Golgi complex of these cells and this phenotype appears to be mediated by deficits in vesicular trafficking (*Ruhl et al., 2019*). We found additional evidence of Syt-17 localization in mDA neurons, including axons, via a Syt-17-EGFP mouse in the GENSAT Brain atlas (*Figure 6—figure supplement 2*; *Heintz, 2004*). We are currently investigating the unknown function of SYT17 in dopaminergic axons. Thus, our APEX2 proteomic dataset elucidates the subcellular localization of proteins encoded by genes within PD risk loci and highlights novel areas for further study of mDA neuronal cell biology.

## Discussion

Our study demonstrates APEX2 labeling and MS-based proteomics of axonal and somatodendritic compartments of mDA neurons in the mouse brain. Thus, APEX2 labeling in acute brain slices provides a general approach for cell type-specific proteomics and/or proximity labeling proteomics in the

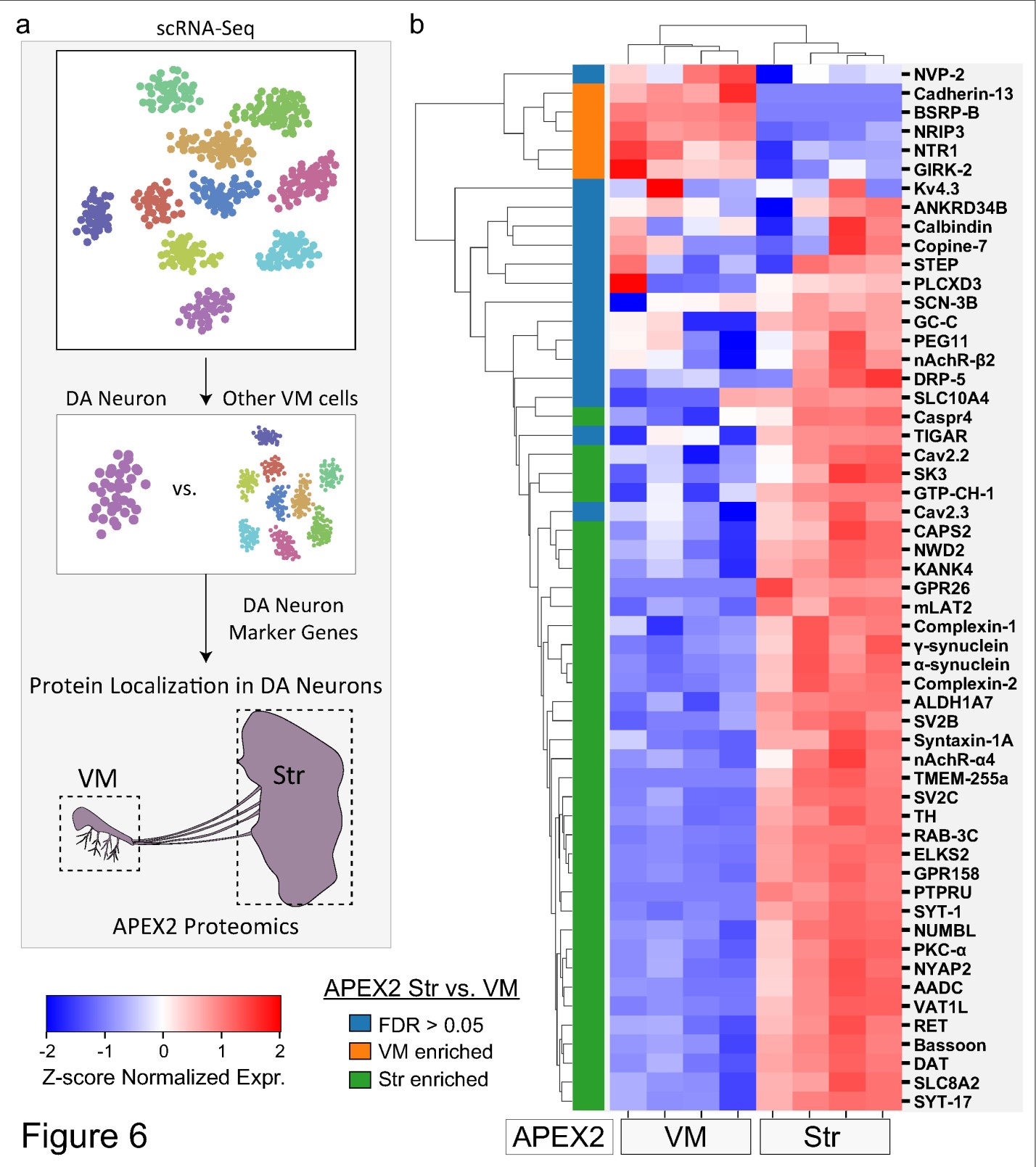

**Figure 6.** Subcellular proteomic analysis of proteins encoded by midbrain dopaminergic (mDA) neuron marker genes. (**a**) Schematic depicting analysis workflow for integrating APEX2 proteomics data with single-cell RNA-sequencing (scRNA-seq) and genetic data. Proteins encoded by top mDA neuron marker genes from mouse scRNA-seq data (*Saunders et al., 2018*) are analyzed for subcellular protein localization in the APEX2 proteomics data. (**b**) Clustered heatmap of Z-scores for abundances of proteins encoded by the top 55 mDA neuron marker genes present in the filtered APEX2 proteomics

*Figure 6 continued on next page*

*Figure 6 continued*

data. Each column represents a biological replicate (*n* = 4) of APEX2⁺ streptavidin pulldown samples from the ventral midbrain (VM) or striatum. The color bar on the left indicates whether a given protein was enriched in the VM (orange) or striatum (green) in differential expression analysis between APEX2⁺ streptavidin pulldown samples (false discovery rate [FDR] <0.05 after Benjamini–Hochberg corrected p values from Welch's *t*-test). See *Figure 6—source data 1* for summary of mDA neuron marker genes and protein abbreviations.

The online version of this article includes the following source data and figure supplement(s) for figure 6:

**Source data 1.** Mouse dopamine (DA) neuronal marker genes, corresponding mouse proteins, and protein abbreviations shown in *Figure 6b*.

**Source data 2.** Unique molecular identifier (UMI) count matrix including high-confidence dopamine (DA) neuron single-cell RNA-sequencing (scRNA-seq) profiles used in this study (data from *Saunders et al., 2018*).

**Figure supplement 1.** Identification of midbrain dopaminergic (mDA) neuron cluster and subclusters for downstream analysis.

**Figure supplement 2.** Midbrain dopaminergic (mDA) neuron enrichment and axonal localization of Syt-17 confirmed via GENSAT mouse JI32.

mouse brain (see also *Dumrongprechachan et al., 2021*), alongside promiscuous biotin ligase (BioID or TurboID)-catalyzed proximity labeling (*Takano et al., 2020*; *Uezu et al., 2016*) and incorporation of non-canonical amino acids via mutant tRNA synthetases (*Alvarez-Castelao et al., 2017*; *Krogager et al., 2018*). Each of these methods has advantages and disadvantages.

The major advantages of APEX2 are speed and efficiency: we were able to capture thousands of proteins from multiple subcellular compartments of a relatively rare neuronal population in individual mice (*Figure 2*). In comparison, BioID and TurboID labeling in the mouse brain typically requires over 7 days and pooling tissue from many mice, precluding the use of individual mice as biological replicates (*Takano et al., 2020*; *Uezu et al., 2016*). The mutant tRNA synthetase methods label all synthesized proteins during 7–21 days of noncanonical amino acid administration, which complicates dynamic studies of specific protein complexes (*Alvarez-Castelao et al., 2017*; *Krogager et al., 2018*). In addition to its high efficiency, the speed of APEX2 labeling enables dynamic studies of protein complexes on a time scale of minutes (*Lobingier et al., 2017*), including during physiological responses in the mouse heart (*Liu et al., 2020*). Future studies might combine APEX2 labeling with optical and electrophysiological slice manipulations to interrogate rapid changes in neuronal physiology at the proteomic level. Beyond neurons, selective expression of APEX2 fusion proteins will broadly facilitate proteomic profiling of organelles and protein–protein interactions within a variety of genetically targeted brain cells.

APEX2 labeling in the mouse brain has several limitations compared to other aforementioned methods. First, the introduction of labeling reagents required preparation of acute brain slices or synaptosomes, which would preclude labeling during long-term behavioral or environmental manipulations (e.g., *Alvarez-Castelao et al., 2017*). Second, the high efficiency of APEX2 may lead to low levels of off-target protein labeling. Although we observed biotin labeling within morphologically intact mDA axons (*Figure 1*) and significant depletion of proteins specific to striatal SPNs and glia (*Figure 2e*), we cannot completely exclude post-synaptic contamination. We propose at least four possible explanations for the detection of proteins with 'postsynapse' annotations in striatal APEX2 samples (*Figure 5—figure supplement 1e*). First, it is possible that many canonical 'postsynaptic' proteins are also localized within mDA axons. For example, we and others have shown that NMDA receptors are present on mDA axons (*Fortin et al., 2012*; *Schmitz et al., 2009*), and most members of the membrane-associated guanylate kinase (PSD-93, PSD-95, SAP-102, SAP-97) and Shank families have been observed in axons and/or nerve terminals (*Aoki et al., 2001*; *Arnold and Clapham, 1999*; *Halbedl et al., 2016*; *Müller et al., 1995*). Indeed, comparison to a previously published dataset revealed that two-thirds of the 'postsynapse' annotated proteins were also present in cultured cortical axons (*Figure 5—figure supplement 1*; *Chuang et al., 2018*). Furthermore, we clearly demonstrated the axonal localization of Kv4.3 and GIRK2, potassium channels typically assumed to be present only within the somatodendritic compartment of mDA neurons (*Figure 4*). Second, it is possible that a small amount of APEX2 is transferred between mDA axons and postsynaptic compartments during slice preparation and synaptosome homogenization. Third, it is possible that BP radicals directly cross damaged axonal and synaptosomal membranes during APEX2 labeling. Finally, it is possible that biotinylated proteins present only in APEX2⁺ samples provide surfaces for nonspecific binding of nonbiotinylated proteins. These mechanisms are not mutually exclusive, but appear to produce similar results in both slice and synaptosome labeling environments.

Although the slice procedure may not be accessible to all laboratories, APEX2 labeling in synaptosomes provides a simple and rapid alternative to access the presynaptic proteome of genetically targeted projection neurons. Somewhat surprisingly, we found that the APEX2 proteome of dopaminergic synaptosomes is largely comparable to that of the entire axonal arbor within the striatum (*Figure 5e*). We suspect this is due to the high density of boutons *en passant* along dopaminergic axons, which would provide a strong representation of the entire axonal proteome upon resealing as synaptosomes. Our data thus highlight the utility of the synaptosome sorting technique developed by Herzog et al. as a complementary approach for study of the presynaptic proteome (*Biesemann et al., 2014*; *Paget-Blanc, 2021*). Future work will determine whether the strong correlation between axonal and synaptosomal proteomes is a unique feature of mDA neurons.

MS-based quantification of proteins provides significant advantages over immunohistochemical methods, especially for fine structures like dopaminergic axons. Suitable antibodies that provide high-quality immunofluorescence in brain tissue are not available for a majority of the proteome, and off-target binding affects the reproducibility and interpretation of research findings (*Rhodes and Trimmer, 2006*; *Weller, 2016*). Many studies rely on overexpression of tagged proteins to establish localization, which can result in mislocalization and altered function. Our study demonstrates proximity labeling of endogenous proteins within subcellular compartments of genetically targeted neurons, and can thus be used for unbiased discovery as well as protein-targeted biochemical experiments.

While they represent the vast majority of neuronal volume and are critically important to DA biology, dopaminergic axons have remained largely inaccessible to proteomic study. Due to the relative immaturity, limited axonal complexity, and incomplete synaptic and hormonal inputs of cultured mDA neurons, we chose to examine the mDA neuronal proteome in native brain tissue of adult mice. Cytoplasmic labeling in dopaminergic axons captured a diverse range of cytosolic and membrane proteins involved in metabolism, protein transport, endolysosomal trafficking, synaptic transmission, and autophagy (*Figures 3 and 4*). Thus, our axonal proteomic dataset should be broadly useful to axonal and neuronal cell biologists.

The axonal enrichment of autophagy-related proteins is of particular importance to mDA neurons, given that autophagy dysfunction is heavily implicated in PD pathophysiology (*Wong and Cuervo, 2010*) and associated with methamphetamine toxicity (*Larsen et al., 2002*). Due to the limitations of bulk striatal tissue-based protein measurements, our previous studies of autophagy in mDA axons were limited to electrochemical measurement of DA release and morphological changes via electron microscopy (*Hernandez et al., 2012*). APEX2 labeling will enable future studies of protein biochemistry in mDA axons, including proteins that are ubiquitously expressed and difficult to resolve using immunohistochemistry.

A majority of mDA neuron-enriched genes encode proteins localized to axons (*Figure 6b*), suggesting that this compartment is central to the identity of mDA neurons. Regardless of expression specificity, the bulk of total protein within mDA neurons is present within their striatal axons (*Figure 2*). These findings may be particularly important for understanding pathogenic mechanisms in PD, wherein the axon is often considered the primary site of degeneration (*Burke and O'Malley, 2013*). Thus, the massive axons of mDA neurons can be considered a double-edged sword: while they are required for DA release to support healthy brain function, they are susceptible to myriad environmental and genetic insults.

We note that the biggest risk factor for idiopathic PD is aging (*Sulzer, 2007*). It is tempting to speculate that the proteomic framework and cytoarchitecture of mDA neurons in the mammalian brain evolved under positive selective pressure related to motor control, reward, and motivation, with little selective pressure related to the organism's lifespan. Comparing the axonal proteome of mDA neurons to that of other neurons spared from degeneration in PD (*Surmeier et al., 2017*) may identify distinguishing features that contribute to increased risk of axonal degeneration in PD. Thus, our study lays a proteomic foundation upon which future studies of neuronal cell biology and PD pathophysiology may build.

## Materials and methods

### Animals

Adult male and female mice (6–12 months old) were used in all experiments. DAT-IRES-Cre mice (*Slc6a3*[tm1.1(cre)Bkmn], JAX #006660, RRID: IMSR_JAX:006660) and Ai9 mice (JAX #007909, RRID: IMSR_JAX:007909) were obtained from Jackson Laboratories (Bar Harbor, ME). Mice were housed on a 12 hr light/dark cycle with food and water available ad libitum. All experiments were conducted according to NIH guidelines and approved by the Institutional Animal Care and Use Committees of Columbia University and the New York State Psychiatric Institute; protocol numbers NYSPI #1584 (Columbia University AABI2605) and NYSPI #1551 (Columbia University AABD8564).

### Plasmid and virus

AAV-CAG-DIO-APEX2NES was a gift from Joshua Sanes (Addgene plasmid #79907; http://n2t.net/addgene:79907; RRID:Addgene 79907). AAV-CAG-DIO-APEX2NES was packaged into an AAV5 vector by Vector BioLabs (Malvern, PA). The final titer of the AAV5-CAG-DIO-APEX2NES preparation was $1 \times 10^{12}$ GC/ml in phosphate-buffered saline (PBS + 5% glycerol). To avoid freeze–thaw, single-use 10 µl aliquots were stored at −80°C.

### Viral injection

All surgical procedures were approved by the Institutional Animal Care and Use Committee and the Department of Comparative Medicine at New York State Psychiatric Institute. Mice were anesthetized with 4% isoflurane. Animals were transferred onto a Kopf Stereotaxic apparatus and maintained under isoflurane anesthesia (1–2%). After hair removal and sterilization of the scalp using chlorhexidine and ethanol, a midline incision was made. Bregma and Lambda coordinates were determined, and minor adjustments in head position were made to match the DV coordinates. Virus was injected at AP −3.2, ML −0.9, and DV −4.4. A small hole was drilled into the skull and 230 nl of virus (see titer above) was injected through a pulled glass pipet using a Nanoject 2000 (Drummond Scientific; 10 pulses of 23 nl). At 5 min after injection, the glass pipet was slowly withdrawn over 5 min. After closing the skin with vicryl sutures, mice received 0.5 ml of 0.9% saline i.p. and were allowed to recover for >1 hr before being returned to their home cages. Animals were housed for at least 3 weeks after injection to allow AAV expression before being experiments.

### Antibodies and reagents

| Name | Manufacturer | Catalog | RRID | Usage |
|---|---|---|---|---|
| Mouse anti-V5 | Thermo Fisher | R960-25 | RRID:AB_2556564 | IHC, 1:1000 WB, 1:1000 |
| Chicken anti-TH | Millipore | AB9702 | RRID:AB_570923 | IHC, 1:500 ICC, 1:1000 |
| Rabbit anti-RFP | Rockland | 600-401-379 | RRID:AB_2209751 | IHC, 1:500 |
| Mouse anti-Kv4.3 | UC Davis/NIH NeuroMab Facility (Antibodies Inc) | K75/41 (75-017) | RRID:AB_2314723 | IHC, 1:1500 |
| Rabbit anti-GIRK2 | Alomone Labs | APC-006 | RRID:AB_2040115 | IHC, 1:1500 |
| Mouse anti-TH | Millipore | MAB318 | RRID:AB_2201528 | WB, 1:2000 |
| Mouse anti-βIII tubulin | Abcam | ab78078 | RRID:AB_2256751 | WB, 1:1000 |
| Mouse anti-Synaptophysin | Agilent | M0776 | RRID:AB_2199013 | WB, 1:500 |
| Goat anti-Chicken IgY (H + L), Alexa Fluor Plus 488 | Thermo Fisher | A-32931TR | RRID:AB_2866499 | IHC/ICC, 1:1000 |
| Goat anti-Mouse IgG (H + L), Alexa Fluor 647 | Thermo Fisher | A-21235 | RRID:AB_2535804 | IHC/ICC, 1:1000 |
| Goat anti-Rabbit IgG (H + L), Alexa Fluor Plus 647 | Thermo Fisher | A32733 | RRID:AB_2633282 | IHC, 1:1000 |

*Continued on next page*

*Continued*

| Name | Manufacturer | Catalog | RRID | Usage |
|------|-------------|---------|------|-------|
| Goat anti-Rabbit IgG (H + L), Alexa Fluor 555 | Thermo Fisher | A-21428 | RRID:AB_2535849 | IHC, 1:750 |
| Streptavidin-AlexaFluor647 | Thermo Fisher | S-21374 | RRID:AB_2336066 | IHC, 1:400 ICC, 1:500 |
| Streptavidin HRP | Vector Laboratories | SA-5014 | RRID:AB_2336510 | WB, 3:10,000 |
| Goat anti-Mouse HRP | Jackson ImmunoResearch | 115-005-003 | RRID:AB_2338447 | WB, 1:5000 |
| Biotin tyramide (Biotin Phenol/BP) | Iris Biotech GmbH | LS3500 | NA | APEX2, 0.5 mM |
| Hydrogen Peroxide ($H_2O_2$) | MilliporeSigma | H1009 | NA | APEX2, 1 mM |

## Acute brain slice preparation

Mice were anesthetized with euthasol and transcardially perfused with 10–15 ml of ice-cold cutting solution (92 mM N-Methyl-D-glucamine/NMDG, 2.5 mM KCl, 30 mM $NaHCO_3$, 20 mM N-2-hydroxyethylpiperazine-N-2-ethane sulfonic acid (HEPES), 1.25 mM $NaH_2PO_4$, 2 mM thiourea, 5 mM sodium ascorbate, 3 mM sodium pyruvate, 10 mM $MgSO_4$, 0.5 mM $CaCl_2$, and 25 mM D-glucose) at pH 7.3–7.4 and saturated with 95% $O_2$/5% $CO_2$. Brains were rapidly extracted and placed into ice-cold cutting solution saturated with 95% $O_2$/5% $CO_2$. For coronal slices, the brain was split roughly in half with a coronal cut at the level of the midhypothalamus (approximately 1.5 mm posterior to Bregma), and the cut surfaces were glued directly to the vibratome stage (Leica VT1000S). For sagittal slices, brains were laterally split in half at the midline and the cut (medial) surfaces were glued to 2% agarose blocks angled at 11°. 300-µm-thick slices were prepared in ice-cold cutting solution continuously saturated with 95% $O_2$/5% $CO_2$. The brainstem and cerebellum were removed from sagittal slices. A typical set of coronal or sagittal slices from a mouse are shown in *Figure 1—figure supplement 2a*.

## Electrophysiological recordings

After preparation in cold cutting solution, slices were transferred to oxygenated normal ACSF containing 125.2 NaCl, 2.5 KCl, 26 $NaHCO_3$, 1.3 $MgCl_2$·$6H_2O$, 2.4 $CaCl_2$, 0.3 $NaH_2PO_4$, 0.3 $KH_2PO_4$, and 10 D-glucose (pH 7.4, 290 ± 5 mOsm) at 34°C and allowed to recover for at least 40 min before the recordings. After recovery, slices were transferred to ACSF ± 0.5 mM BP and incubated for 1 hr at room temperature. Electrophysiological recordings were performed on an upright Olympus BX50WI (Olympus, Tokyo, Japan) microscope equipped with a ×40 water immersion objective, differential interference contrast (DIC) optics and an infrared video camera. Slices were transferred to a recording chamber and maintained under perfusion with normal ACSF (1.5–2 ml/min) at 34°C. All recorded dopaminergic neurons were located in the substantia nigra pars compacta (SNc) and identified by larger somatic size than neighboring neurons. Patch pipettes (3–5 MΩ) were pulled using P-97 puller (Sutter instruments, Novato, CA) and filled with internal solutions contained (in mM): 115 K-gluconate, 10 HEPES, 2 $MgCl_2$, 20 KCl, 2 MgATP, 1 $Na_2$-ATP, and 0.3 GTP (pH = 7.3; 280 ± 5 mOsm).

Cell-attached patch clamp recordings were performed with a MultiClamp 700B amplifier (Molecular Devices, Forster City, CA) and digitized at 10 kHz with InstruTECH ITC-18 (HEKA, Holliston, MA). Once obtaining stable patch configuration, spontaneous firing was recorded for 3 min. Data were acquired using WINWCP software (developed by John Dempster, University of Strathclyde, UK) and analyzed using Clampfit (Molecular Devices) and Igor Pro (Wavemetrics, Lake Oswego, OR).

## APEX2 biotinylation in brain slices

After preparation in cold cutting solution, slices were transferred to jars containing 70 ml of aCSF (125.2 mM NaCl, 2.5 mM KCl, 26 mM $NaHCO_3$, 1.3 mM $MgCl_2$·$6H_2O$, 2.4 mM $CaCl_2$, 0.3 mM $NaH_2PO_4$, 0.3 mM $KH_2PO_4$, 10 mM D-glucose) supplemented with 0.5 mM BP and 1 µM tetrodotoxin. aCSF was continuously saturated with 95% $O_2$/5% $CO_2$. Slices were allowed to recover for 60 min at room temperature. After recovery, APEX2 labeling was initiated by the addition of 1 mM $H_2O_2$ to aCSF at room temperature. We tested labeling periods of 1–5 min (*Figure 1—figure supplement 2b–d*) and found 3 min to be sufficient for downstream applications. To quench labeling, slices were rapidly transferred to a separate jar containing 50 ml of quenching aCSF (aCSF supplemented with 10 mM Trolox,

20 mM sodium ascorbate, and 10 mM NaN$_3$). After 5 min rest at room temperature, slices were either transferred to fixative solution for downstream immunostaining, or transferred to ice-cold quenching aCSF for rapid dissection and downstream western blotting/proteomics.

## Synaptosome preparation and APEX2 biotinylation

Synaptosomes were prepared using standard procedures (*Gray and Whittaker, 1962*). Mice were sacrificed by cervical dislocation, after which forebrains were rapidly dissected and placed in 10 volumes of ice-cold buffer consisting of 0.32 M sucrose, 4 mM HEPES, pH 7.4, and protease inhibitors (cOmplete ethylenediaminetetraacetic acid (EDTA)-free protease inhibitor, Roche). Tissue was homogenized on ice in a glass–glass dounce homogenizer with 10 gentle strokes of loose and tight clearance pestles. All subsequent purification steps were performed on ice or at 4°C unless otherwise specified. The homogenate was centrifuged at 1000 × *g* (Eppendorf 5424 R) for 10 min to remove nuclei and cellular debris, yielding a P1 pellet and an S1 supernatant. The S1 supernatant was further centrifuged at 10,000 × *g* for 15 min to obtain the crude synaptosome pellet (P2). The P2 pellet was resuspended in sucrose buffer, incubated for 5 min on ice, and recentrifuged at 10,000 × *g* for 15 min. The washed P2 pellet was resuspended in PBS with 0.5 mM BP and incubated for 30 min at room temperature. For immunostaining experiments, the 30-min incubation in PBS + BP occurred on poly-lysine coated coverslips to allow synaptosome sedimentation and adherence. Unbound synaptosomes were removed by several brief washes in PBS + BP prior to APEX2 labeling. APEX2 labeling was initiated by the addition of 0.5× volumes of 2 mM H$_2$O$_2$ (1 mM final). After 60 s, labeling was quenched by addition 4× volumes of ice-cold PBS + 12.5 mM Trolox, 25 mM sodium ascorbate, and 12.5 mM NaN$_3$ (10, 20, and 10 mM final, respectively). For immunostaining, synaptosomes were washed several more times in quenching PBS over 5 min, followed by addition of fixative solution. For western blotting and proteomics, synaptosomes were collected by recentrifugation at 10,000 × *g* for 15 min, flash frozen on liquid nitrogen, and stored at −80°C until further use.

## Histology and immunofluorescence

For initial characterization of AAV5-CAG-DIO-APEX2NES specificity, mice were anesthetized with euthasol and transcardially perfused with ~15 ml of 0.9% saline followed by 40–50 ml of ice-cold 4% paraformaldehyde (PFA) in 0.1 M phosphate buffer (PB), pH 7.4. Brains were postfixed in 4% PFA in 0.1 M PB for 6–12 hr at 4°C, washed three times in PBS, and sectioned at 50 µm on a Leica VT1000S vibratome. Sections were placed in cryoprotectant solution (30% ethylene glycol, 30% glycerol, 0.1 M PB, pH7.4) and stored at −20°C until further use.

Sections were removed from cryoprotectant solution and washed three times in tris-buffered saline (TBS) at room temperature. Sections were then permeabilized in TBS + 0.3% Triton X-100 for 1 hr at room temperature, followed by blocking in TBS + 10% normal goat serum (NGS) and 0.3% Triton X-100 for 1.5 hr at room temperature. Sections were then directly transferred to a prechilled solution containing primary antibodies in TBS +2% NGS + 0.1% Triton X-100 and incubated overnight at 4°C. Sections were washed in TBS five times over an hour at room temperature. Sections were incubated in a solution containing secondary antibodies in TBS + 2% NGS + 0.1% Triton X-100 at room temperature for 1.5 hr, followed by four washes in TBS + T over 45 min at room temperature. Following four additional washes in TBS, sections were slide mounted and coverslipped with Fluoromount G (Southern Biotech). See *Antibodies and reagents* for a complete list of antibodies and concentrations used in this study.

For immunostaining of acute brain slices after APEX2 labeling, slices were transferred to ice-cold 4% PFA in 0.1 M PB + 4% sucrose and fixed overnight at 4°C. To remove lipids and enhance antibody penetration, fixed slices were transferred to CUBIC solution 1A, consisting of 10% wt Triton X-100, 5% wt NNNN-tetrakis (2-HP) ethylenediamine, 10% wt urea, and 25 mM NaCl (*Susaki and Ueda, 2016*). Slices were blocked in TBS + 10% NGS and 0.3% Triton X-100 for 24 hr at room temperature and then transferred to TBS + 2% NGS and 0.3% Triton X-100 supplemented with primary antibodies. After 72-hr incubation at 4°C with primary antibodies, slices were washed five times over 10 hr in TBS. Slices were incubated for 24 hr at room temperature in TBS + 2% NGS + 0.3% Triton X-100 supplemented with secondary antibodies and fluorophore-conjugated streptavidin. After five washes in TBS over 10 hr, sections were slide mounted and coverslipped with Fluoromount G. See *Antibodies and reagents* for a complete list of antibodies and concentrations used in this study.

For immunostaining of synaptosomes after APEX2 labeling, synaptosomes adhered to poly-lysine coverslips were fixed with 4% PFA in 0.1 M PB + 4% sucrose for 10 min at room temperature. After several washes in PBS, synaptosomes were incubated with PBS + 0.1 M glycine + 0.05% Tween-20 for 15 min at room temperature. After blocking/permeabilization with PBS + 10% NGS + 0.2% Tween-20 for 1 hr at room temperature, synaptosomes were incubated with primary antibodies in PBS + 2% NGS + 0.1% Tween-20 at 4°C overnight. After three washes in PBS, synaptosomes were incubated in secondary antibodies in PBS + 2% NGS + 0.1% Tween-20 for 1 hr at room temperature. After three more washes in PBS, coverslips were stored and imaged in Fluoromount G. See *Antibodies and reagents* for a complete list of antibodies and concentrations used in this study.

## Tissue lysis and protein processing

Capture and processing of biotinylated proteins was conducted as previously described (*Kalocsay, 2019*; *Liu et al., 2020*) with only minor modifications. Immediately after dissection in ice-cold quenching aCSF, tissues were flash frozen in liquid nitrogen and stored at −80°C until further use. Frozen tissues or synaptosome pellets were homogenized on ice in a glass dounce homogenizer (Sigma D9063) with 30 strokes of both A and B pestles. Lysis was in 0.75 ml of ice-cold tissue lysis buffer, consisting of 50 mM Tris, pH 8.0, 150 mM NaCl, 10 mM EDTA, 1% Triton X-100, 5 mM Trolox, 10 mM sodium ascorbate, 10 mM sodium azide, and 1× EDTA-free protease inhibitors (Roche). After addition of 39 µl of 10% SDS (final concentration 0.5%), lysates were rotated for 15 min at 4°C. Lysates were clarified by centrifugation at 21,130 × *g* for 10 min at 4°C. Supernatants were transferred to a new prechilled Eppendorf tube for trichloroacetic acid (TCA) precipitation (for MS) or stored at −80°C (for western blotting).

Proteins were precipitated from lysates by the addition of an equal volume of ice-cold 55% TCA. Samples were incubated on ice for 15 min, followed by centrifugation at 21,130 × *g* for 10 min at 4°C. Protein pellets were resuspended in 1 ml of acetone prechilled to −20 °C and recentrifuged as before. Pellets were resuspended and recentrifuged another three times in 1 ml of acetone prechilled to −20°C, for a total of four washes. Residual acetone was removed, and protein pellets were resuspended in Urea Dissolve Buffer (8 M urea, 1% SDS, 100 mM sodium phosphate, pH 8, 100 mM NH$_4$HCO$_3$). Dissolution of pellets was facilitated by water both sonication for 10 min followed by gentle agitation on an orbital shaker for 1 hr at room temperature. Removal of residual TCA was confirmed by checking that the pH ~8.0. In some cases, a small aliquot (5%) of the resuspended protein was flash-frozen and stored at −80°C. 1/49 the volume of 500 mM TCEP (Sigma, cat. #646547) and 1/19 the volume of freshly prepared 400 mM iodoacetamide (Thermo Fisher, cat. #90034) in 50 mM NH$_4$HCO$_3$ was added to the protein resuspension for disulfide reduction and cysteine alkylation at final concentrations of 10 mM TCEP and 20 mM iodoacetamide. The suspension was vortexed and incubated in the dark for 25 min at room temperature. Alkylation was quenched by addition of 1/19 the volume of 1 M dithiothreitol (DTT) to reach 50 mM DTT. Samples were diluted with 0.87× volumes of H$_2$O to reach a final concentration of 4 M urea and 0.5% SDS.

## Capture of biotinylated proteins for MS

Streptavidin magnetic beads (Thermo Fisher #88817) were resuspended and washed three times in Urea Detergent Wash Buffer (4 M Urea, 0.5% SDS, 100 mM sodium phosphate, pH 8) for at least 10 min at 4°C. After washing, streptavidin beads were resuspended in ice-cold Urea Detergent Wash Buffer and 50 µl containing 0.5 mg of beads was added to each sample. Proteins were incubated with streptavidin beads overnight on a rotor at 4°C. After 14–18 hr, the unbound supernatant was discarded, and beads were resuspended in 1 ml of Urea Detergent Wash Buffer and transferred to a new tube. Beads were washed three times for 5–10 min in 1 ml of Urea Detergent Wash Buffer at room temperature. After the third wash, beads were resuspended in 1 ml of Urea Wash Buffer (4 M urea, 100 mM sodium phosphate, pH 8) and transferred to a new tube. After three 5–10 min washes in 1 ml of Urea Wash Buffer at room temperature, beads were resuspended in 200 µl of Urea Wash Buffer and transferred to a new tube. A 10 µl aliquot (5%) was transferred to a separate tube for western blotting, and the remaining 190 µl of buffer were removed on a magnetic stand. Beads were flash frozen and stored at −80°C.

## Western blotting

The protein concentration of frozen tissue lysates was determined using the BCA assay (Pierce, Thermo Fisher, cat. #23225) and diluted with 4× lithium dodecyl sulfate (LDS) sample buffer (Thermo Fisher, cat. #NP0007) supplemented with 20 mM DTT and boiled for 5 min at 95°C. Frozen streptavidin beads were resuspended in ~20 µl of 1× LDS sample buffer supplemented with 20 mM DTT and 2 mM biotin. Samples were boiled for 5 min at 95°C to elute biotinylated proteins. Beads were immediately placed immediately onto a magnetic rack and the entire sample was immediately loaded into 10% Bis–Tris polyacrylamide gels (Invitrogen, Thermo Fisher, cat. #NP0303BOX) and transferred to Poly-vinylidene Fluoride(PVDF) membranes (Immobilon-P, MilliporeSigma, cat. #IPVH00010). Membranes were initially washed for 15 min in TBST (1× TBS + 0.1% Tween 20), blocked for an hour in 5% bovine serum albumin (BSA)/TBST, and incubated overnight at 4°C with primary antibody in 5% bovine serum albumin/TBST overnight. After primary incubation, membranes were washed three times in TBST prior to incubation with streptavidin–HRP or HRP-conjugated secondary antibody in 2.5% BSA/TBST for 1 hr at room temperature. After secondary incubation, membranes were washed three times in TBST. Signal was developed using Immobilon enhanced chemiluminescent substrate (Millipore, cat. #WBKLS0500) and imaged on an Azure Biosystems C600 system.

## On bead digestion and (LC–MS/MS)

Proteins bounded streptavidin beads were resuspended in 200 µl of digestion buffer (1 M urea, 100 mM EPPS, pH 8.5, 4% acetonitrile) and digested with 2 µg of trypsin/LysC mix overnight at 37°C. The next day, digested peptides were collected in a new microfuge tube and digestion was stopped by the addition of 1% TFA (final vol/vol), followed by centrifugation at 14,000 × g for 10 min at room temperature. Cleared digested peptides were desalted on an SDB-RP Stage-Tip and dried in a speed-vac. Dried peptides were dissolved in 3% acetonitrile/0.1% formic acid. Desalted peptides (300–500 ng) were injected onto an EASY-Spray PepMap RSLC C18 50 cm × 75 µm column (Thermo Scientific), which was coupled to the Orbitrap Fusion Tribrid mass spectrometer (Thermo Scientific). Peptides were eluted with a nonlinear 120 min gradient of 5–30% buffer B (0.1% [vol/vol] formic acid, 100% acetonitrile) at a flow rate of 250 nl/min. The column temperature was maintained at a constant 50°C during all experiments.

Samples were run on the Orbitrap Fusion Tribrid mass spectrometer with a DIA method for peptide MS/MS analysis (*Bruderer et al., 2015*). Survey scans of peptide precursors were performed from 350 to 1200 *m/z* at 120 K FWHM resolution (at 200 *m/z*) with a $1 \times 10^6$ ion count target and a maximum injection time of 60 ms. After a survey scan, 26 *m/z* DIA segments acquired at from 200 to 2000 *m/z* at 60 K FWHM resolution (at 200 *m/z*) with a $1 \times 10^6$ ion count target and a maximum injection time of 118 ms. HCD fragmentation was applied with 27% collision energy and resulting fragments were detected using the rapid scan rate in the Orbitrap. The spectra were recorded in profile mode. All data have been deposited to the ProteomeXchange Consortium via the PRIDE (*Perez-Riverol et al., 2019*) partner repository with the dataset identifier PXD026229.

## Raw MS data processing

DIA data were analyzed with directDIA 2.0, a spectral library-free analysis pipeline featured in Spectronaut Pulsar X software (Biognosys AG). The default settings were used for targeted analysis of DIA data in Spectronaut except the decoy generation was set to 'mutated'. False discovery rate (FDR) was estimated using the mProphet approach (*Reiter et al., 2011*) and set to 1% at peptide precursor level and at 1% at protein level. For peptides and proteins that were not detected in a given sample (directDIA output as 'Filtered'), the intensity was set to 0 for downstream analysis.

## Proteomic differential expression analysis and filtering

Total intensity normalized, log$_2$-transformed protein abundances were used for visualization, clustering, and all differential abundance analyses. Log-normalized protein abundances within each sample are given by:

$$log_2 \left( \frac{X_i}{\sum X_n} \times 10^6 + 1 \right)$$

where $X_i$ is the raw intensity for protein $i$, and the total intensity is calculated as the summed raw intensity for all proteins ($\sum X_n$). Differential abundance testing consisted of a Welch's (unequal variance) $t$-test with Benjamini–Hochberg procedure to control the FDR.

A graphical summary of the initial filtering for VM and striatum APEX2 proteomics data is shown in *Figure 3—figure supplement 1d*. Most filters were based on the APEX2 proteomics data alone, with some additional filters based on mDA neuron scRNA-seq data derived from *Saunders et al., 2018* (see below). VM proteins were filtered as follows:

1. Proteins meeting statistically significant (FDR <0.05) enrichment in APEX2$^+$ vs. APEX2$^-$ (Control) differential expression were retained.
2. Proteins were retained if they met either one of the two following conditions (a OR b):
   a. Mean mDA neuron scRNA-seq expression above the lower bound (mean − standard deviation).
   b. Statistically significant (FDR <0.05) enrichment in APEX2$^+$ vs. APEX2$^-$ (Control) differential expression in all three regions (VM, MFB, and Str).
3. Proteins were removed if they were encoded by genes with very low mDA neuron specificity in scRNA-seq data (Mann–Whitney $U$-test comparing mDA neurons vs. all other midbrain cells).

Striatum proteins were filtered as follows:

1. Proteins meeting statistically significant (FDR <0.05) enrichment in APEX2$^+$ vs. APEX2$^-$ (Control) differential expression were retained.
2. Proteins were retained if they met any of the following conditions (a OR b OR c):
   a. Statistically significant enrichment (FDR <0.05) in APEX2$^+$ vs. APEX2$^-$ (Control) differential expression for either VM or MFB.
   b. log$_2$ fold change >1 in APEX2$^+$ vs. APEX2$^-$ (Control) comparisons for both VM and MFB samples.
   c. High mean mDA neuron scRNA-seq expression (above the mean plus standard deviation) and mDA neuron specificity (Mann–Whitney $U$-test comparing mDA neurons vs. all other midbrain cells or vs. all striatal cells).
3. Proteins were retained if they met either one of the two following conditions (a OR b):
   a. Mean mDA neuron scRNA-seq expression above the lower bound (mean minus standard deviation).
   b. Statistically significant (FDR <0.05) enrichment in APEX2$^+$ vs. APEX2$^-$ (Control) differential expression in all three regions (VM, MFB, and Str).
4. Proteins were removed if they were encoded by genes with very low mDA neuron specificity in scRNA-seq data (Mann–Whitney $U$-test comparing mDA neurons vs. all other midbrain cells or vs. all striatal cells).

## GO analysis

For all GO analyses, a single list of unique genes encoding the corresponding proteins was used (i.e., a single gene entry was used when multiple protein isoforms in the list were encoded by a single gene). For peptides/protein groups mapped to multiple, homologous proteins, the first gene entry as determined by Spectronaut default settings was used. The VM vs. striatum GO analysis shown in *Figure 3d* was conducted using web-based Enrichr (*Xie et al., 2021*) with 2018 GO Terms for Cellular Component, Biological Process, and Molecular Function (*Ashburner et al., 2000*; *The Gene Ontology Consortium, 2021*). Enrichr was also used for subcellular compartments analysis shown in *Figure 3—figure supplement 3a, b*. Additional targeted GO analysis shown in *Figure 3—figure supplement 3c, d* was conducted manually using the *SciPy* implementation of the hypergeometric test. Nuclear-related ontologies were obtained from COMPARTMENTS (*Binder et al., 2014*) and mitochondrial localization ontologies were obtained from MitoCarta 3.0 (*Rath et al., 2021*). The synaptic GO analysis shown in *Figure 5* and *Figure 5—figure supplement 1* was conducted using SynGO (*Koopmans et al., 2019*).

## Image acquisition and analysis

Imaging of 50 µm sections from perfusion-fixed brain was conducted on a Nikon Ti2 Eclipse epifluorescence microscope or on a Leica SP8 scanning confocal microscope. To confirm the specificity of V5 expression in mDA neurons, tile scan epifluorescence images of the entire VM were collected at 2–3 $z$-planes per section. V5-positive neurons were first identified using only the V5 channel and their

somas were segmented as ROIs. Each V5-positive neuronal ROI was subsequently scored for tdTomato and TH expression. Neurons within both the SNc and ventral tegmental area were quantified. High-resolution (×60/1.4 NA) confocal images of striatal sections confirmed that V5 (APEX2), tdTomato, and TH were localized exclusively within dopaminergic axons. 300-μm-thick brain slices were imaged only by confocal microscopy. To assess biotin labeling throughout the slice, Z-stacks spanning the entire slice depth were acquired at 4.18 μm intervals using a ×20/0.4 NA objective.

## scRNA-seq analysis

For comparison to scRNA-seq, we obtained Drop-seq count matrices for substantia nigra and striatum from GSE116470 (DropViz, *Saunders et al., 2018*). To identify DA neurons, we first performed unsupervised clustering on the substantia nigra count matrices using the Phenograph (*Levine et al., 2015*) implementation of Louvain community detection after selection of highly variable genes and construction of a *k*-nearest neighbors graph as described previously (*Levitin et al., 2019*). We identified a single cluster with statistically significant coenrichment of DA neuron markers such as *Th* and *Slc6a3* based on the binomial test for expression specificity (*Shekhar et al., 2016*) as shown in the UMAP (*Becht et al., 2018*) embedding in *Figure 6—figure supplement 1a*. After subclustering the putative DA neurons using the methods described above, we identified a small subcluster with statistical enrichment of astrocyte markers such as *Agt*, *Gja1*, *Glul*, and *Slc1a3*. We discarded this subcluster as likely astrocyte contamination and removed all remaining cells with fewer than 1000 unique transcripts detected to produce a count matrix of high-confidence DA neuron profiles. We subclustered these profiles to identify five transcriptionally distinct DA neuron subsets with markers determined using the binomial test shown in *Figure 6—figure supplement 1b*.

We used these high-confidence profiles to identify genes with enrichment in DA neurons compared to all midbrain cells by differential expression analysis as described in *Szabo et al., 2019* with minor modifications. Briefly, to perform differential expression analysis between two groups of cells, we randomly subsampled the data so that both groups are represented by the same number of cells. Next, we randomly subsampled the detected transcripts so that both groups have the same average number of transcripts per cell. Finally, we normalized the two subsampled count matrices using *scran* (*Lun et al., 2016*) and analyzed differential expression for each gene using the SciPy implementation of the Mann–Whitney *U*-test. We corrected the resulting p values for false discovery using the Benjamini–Hochberg procedure as implemented in the *statsmodels* package in Python. We used differential expression analyses between high-confidence DA neurons and the remaining cells in the midbrain to select genes with >eightfold specificity for expression in mDA neurons ($\log_2$ FC >3 and FDR <0.01).

Analysis of Proteins Encoded by mDA Neuron Marker Genes mDA neuron marker genes (genes specific to mouse DA neurons) were identified using scRNA-seq data as described above. Out of 64 genes with >eightfold specificity for expression in mDA neurons, 55 corresponding proteins were present in the filtered APEX2 proteomics data. See *Figure 6—source data 1* for complete summary of mouse mDA neuronal marker genes, corresponding mouse proteins, and protein abbreviations shown in *Figure 6b*.

## Visualization and statistical analysis

Cartoon graphics (e.g., *Figures 3d and 5f*) were created in Adobe Illustrator 24.3 (Adobe, Inc) with additional illustrations from BioRender (https://biorender.com/). Proteins were selected for display on this basis of inclusion in significantly over-represented GO Terms, with additional proteins selected based on manual curation of the relevant literature. Unless otherwise noted, all proteins displayed were present in the filtered APEX2 proteomics data. Protein abbreviations and corresponding full protein names are provided as source data for each respective figure.

Unless otherwise noted, all statistical analysis and data visualization were conducted in Python using *SciPy*, *Matplotlib*, and *Seaborn* packages. For visualization of proteomics data (log–log abundance plots, Z-scores, clustered heatmaps, etc.), total intensity normalized protein abundances were $\log_2$ transformed after adding 1. The total intensity normalized, $\log_2$ transformed protein intensities are generally referred to as $\log_2$ protein abundance, as specified in figure captions. For clustered heatmaps, Z-scores of $\log_2$ protein abundances were first calculated using the *zscore* function within the *SciPy Stats* module, after which the row and column clustering was calculated using the *linkage*

function (metric = 'Euclidean', method = 'average') within *fastcluster 1.2.3* (*Müllner, 2013*) and passed to *Seaborn clustermap*.

## Materials availability

There are restrictions to the availability of AAV5-CAG-DIO-APEX2NES virus due to limited production size. The exact plasmid used for production of this virus (AAV-CAG-DIO-APEX2NES, Addgene plasmid #79907) can be ordered from Addgene and/or sent directly to Vector BioLabs for further production.

## Acknowledgements

This work was conducted in collaboration with the Proteomics Shared Resource within the Herbert Irving Comprehensive Cancer Center at Columbia University Irving Medical Center (NIH Grant 2P30 CA013696-45). This research was funded in part by Aligning Science Across Parkinson's [ASAP-000375] (DS and PAS) through the Michael J Fox Foundation for Parkinson's Research (MJFF). For the purpose of open access, the author has applied a CC BY public copyright license to all Author Accepted Manuscripts arising from this submission. This work was supported by the JPB Foundation (DS). This work was supported by NIH grants F30 DA047775-03 (BDH), R01 NS095435 (DS), R01 DA007418 (DS), and R01 MH122470 (DS). We would like to thank Vanessa Morales for assistance with animal colony management.

## Additional information

### Funding

| Funder | Grant reference number | Author |
|---|---|---|
| National Institutes of Health | F30DA047775 | Benjamin D Hobson |
| National Institutes of Health | R01NS095435 | David Sulzer |
| National Institutes of Health | R01DA007418 | David Sulzer |
| National Institutes of Health | R01MH122470 | David Sulzer |
| Aligning Science Across Parkinson's | ASAP-000375 | David Sulzer Peter Sims |

The funders had no role in study design, data collection, and interpretation, or the decision to submit the work for publication.

### Author contributions

Benjamin D Hobson, Conceptualization, Formal analysis, Investigation, Methodology, Supervision, Visualization, Writing – original draft, Writing – review and editing; Se Joon Choi, Formal analysis, Investigation; Eugene V Mosharov, Investigation, Methodology; Rajesh K Soni, Investigation, Methodology, Writing – review and editing; David Sulzer, Funding acquisition, Resources, Supervision, Writing – review and editing; Peter A Sims, Formal analysis, Funding acquisition, Supervision, Writing – original draft, Writing – review and editing

### Author ORCIDs

Benjamin D Hobson http://orcid.org/0000-0002-2745-5318
Rajesh K Soni http://orcid.org/0000-0002-4556-4358
David Sulzer http://orcid.org/0000-0001-7632-0439
Peter A Sims http://orcid.org/0000-0002-3921-4837

### Ethics

All experiments were conducted according to NIH guidelines and approved by the Institutional Animal Care and Use Committees of Columbia University and the New York State Psychiatric Institute.

Protocol numbers are NYSPI #1584 (Columbia University AABI2605) and NYSPI #1551 (Columbia University AABD8564).

### Decision letter and Author response
Decision letter https://doi.org/10.7554/eLife.70921.sa1
Author response https://doi.org/10.7554/eLife.70921.sa2

## Additional files

### Supplementary files
• Transparent reporting form

### Data availability
The mass spectrometry proteomics data have been deposited to the ProteomeXchange Consortium via the PRIDE (Perez-Riverol et al., 2019) partner repository with the dataset identifier PXD026229. Raw label-free quantification intensity values for proteomics data can be found in Figure 2 - source data 2. The scRNA-seq data analyzed are publicly available as GSE116470 (Saunders et al., 2018). High confidence DA neuron profiles used in this study are reported in Figure 5 - source data 3.

The following dataset was generated:

| Author(s) | Year | Dataset title | Dataset URL | Database and Identifier |
|---|---|---|---|---|
| Hobson BD, Sims PA | 2021 | Axonal and somatodendritic proteomes of dopamine neurons in the mouse brain | http://proteomecentral.proteomexchange.org/cgi/GetDataset?ID=PXD026229 | ProteomeXchange, PXD026229 |

The following previously published dataset was used:

| Author(s) | Year | Dataset title | Dataset URL | Database and Identifier |
|---|---|---|---|---|
| Saunders A, McCarroll S | 2018 | A Single-Cell Atlas of Cell Types, States, and Other Transcriptional Patterns from Nine Regions of the Adult Mouse Brain | https://www.ncbi.nlm.nih.gov/geo/query/acc.cgi?acc=GSE116470 | NCBI Gene Expression Omnibus, GSE116470 |

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
