## [Editor Report]

In this work, the authors provide a useful compendium of proteins labeled within dopaminergic cells using a novel approach. Novel viral approaches were developed to rapidly biotinylate proteins in dopaminergic neurons in oriented sections of brain whereby circuits can be spatially parsed for proteomic dissection. In addition to providing a useful new database of proteins for investigators interested in this circuit, the results also provide a more general approach to examining a compartment proteome in neurons and what might be expected in that analysis in an unbiased way not previously envisaged.

---

## [Decision Letter]

**Decision letter after peer review:**

Thank you for submitting your article "Subcellular proteomics of dopamine neurons in the mouse brain reveals axonal enrichment of proteins encoded by Parkinson's disease-linked genes" for consideration by *eLife*. Your article has been reviewed by 3 peer reviewers, one of whom is a member of our Board of Reviewing Editors, and the evaluation has been overseen by Gary Westbrook as the Senior Editor. The reviewers have opted to remain anonymous. The reviewers have discussed their reviews with one another, and the Reviewing Editor has drafted this to help you prepare a revised submission.

Essential revisions:

1) All reviewers and the Reviewing editor agreed that additional protein validation is needed, potentially with orthogonal antibody staining approaches. While the synaptosome preps presented are supportive of the BioIDs, and may facilitate candidate selection, the synaptosome preps do not obviate the need for an orthogonal approach. In protein validation, the target could be evaluated in other circuits as well which would help provide insight into how unique the localization in the dopaminergic neurons really is compared to other types of neurons.

2) All reviewers and the Reviewing editor were confused by the steps taken by the authors to normalize the data. Two of the reviewer's thought that it is critical to normalize the abundance of biotinylated proteins from one lysate to the next to the amount of APEX2 enzyme present in the lysate, or provide a justification of why more peroxidase expression would not be expected to label more distinct protein substrates and label protein substrates to a higher level? Normalization methods of all mass spectrometry data should be included briefly in all relevant figure legends, as well as a clear work-flow in the Methods section.

3) Two reviewers thought the comparative analysis of the PD GWAS data with the proteome IDs was imprecise at best, and potentially misleading. Some specific concerns include the conflation of strong recessive, dominant, and risk factor variants with potentially incorrect gene assignments, arbitrary inclusion of some genes and not others, overly relaxed false-discovery rates, and heavy implicit bias. All reviewers and Reviewing Editor agreed that the manuscript would be more focused without these experiments and the related claims about the heritable aspects of PD and PD-associated diseases.

*Reviewer #1 (Recommendations for the authors):*

The following suggestions for improvement are provided per figure for clarity, although some may be relevant more globally;

Figure 1 is generally showing that the technique works in vivo and is good. One small question is with 1d, where it appears that the V5APEX2 expression is higher in the striatum than in midbrain. Was this consistently seen and might it influence detection sensitivity later in the paper? It would be important to add a reference protein for loading to 1d and quantify relative expression in the two regions.

Figure 2 is the main results figure and again largely makes sense. Here, I would again like to see an evaluation of the amount of V5-APEX2 in each region relative to control, so we can understand the apparent discrepancy between number of proteins detected and % biotinylation between ventral midbrain and striatum. It is notable here that the control is different from figure 1, which was -H_2_O_2_, vs no APEX2. It would therefore be important to include validation experiments using immunoblotting and also include the controls from figure 1 to be sure that these are dependent on APEX2 activity.

Figure 3 is generally fine apart from the discrepancy in FDR p values selected for cutoffs. FDR<0.15 is too liberal, especially given that log fold cutoffs appear not to have been applied and that t-tests were used without evaluation of normal distributions that would be difficult from low n of samples. Again some validation of key results is needed.

Figure 4 particularly needs validation of proteins that are expected to be post-synaptic. I am sure that the literature distinctions between pre and post synaptic are less rigid than might be inferred, but some evaluation of accuracy of this separation is needed.

Figure 5 is the figure that has most problems. Figure 5a is a schematic and 5b tells us that there is partial agreement between proteome and scRNA-seq, which is to be expected. But, 5c, has little informational quality for multiple reasons. For GWAS SNPS, the selection of nearest gene to lead SNP is only true in some uncertain proportion of loci so whether INPP5F is the gene at the Chr10 locus that includes BAG3 and RGS10 is impossible to evaluate. For the smaller set of Mendelian loci, whether we should combine PD and atypical and dystonia is hard to evaluate. At the same time, there are multiple loci that are not in this set – LRRK2 and GBA being very obvious. So at best, this dataset says that some PD genes are in dopamine neurons, which is unsurprising, but not all PD genes are dopamine neuronal. The real problem here is in decision of numerator and denominator. For other tools more widely used in GWAS (FUMA, MAGMA etc) the test set is all candidate genes vs all expressed in a given cell type. Here, the authors compare a set they detect in the cells and then look for cell body vs axon enrichment, which is fundamentally less precise or informative. Authors should look for enrichment within each dataset for all Mendelian PD genes or all GWAs hits, which should include all reasonable candidates within LD-defined bounds of each locus. Even if this turns out to be more than chance, the authors must discuss limitations – DJ-1 is found here but very notably not PINK1/parkin. Such patterns might easily be explained by chance ordering of proteins in Str vs VM and should be discussed adequately.

*Reviewer #2 (Recommendations for the authors):*

1. When doing streptavidin-dependent pulldowns of biotinylated proteins the authors write 'immunoprecipitation'. This is not correct since no antibodies are used. The authors should be technically correct and talk about 'pull-downs' or 'streptavidin-dependent purification of biotinylated proteins.' I realize most readers will understand what they are trying to say, but for such a technically excellent paper I think the authors shouldn't use wrong terminology.

2. In some instances I found the figures to be overly 'busy'. For example, Figure 3d is very busy with a great deal of speculation included about protein function. No studies here actually test biology of the candidates identified by mass spectrometry. The faith (and I label it as such) placed in GO analysis by the authors is not justified. On the other hand, I applaud the authors for attempting to place some of the identified proteins in biological context. What I'd prefer to see is a bit of a disclaimer about the robustness of GO analysis and also include more statements about how this is a discovery approach that will require many follow-up studies to elucidate protein function.

*Reviewer #3 (Recommendations for the authors):*

The compendium seems rigorous and potentially useful to the field in understanding TH neurons in the SNpc.

1. I do not understand why biotinlyated protein abundance in the different preparations from the different compartments are not normalized to APEX2 abundance in the relative quantifications. Overall, the process of normalization was unclear, including how differing AAV transduction efficiency is factored into calculated biotinylated proteins. Unless i am fundamentally mistaken, APEX2 labeling occurs so quickly that the relative abundance of enzyme will influence not only the number of proteins that the mass spectrometry analysis can identify, but also the relative abundance when comparing one preparation of protein to the next.

2. Along those lines, since there are simply more proteins IDed in the striatum, with a larger 'interactome' afforded by more substrate material, and with higher possible proportional abundance of APEX2 , it makes sense that there are more PD- GWAS genes identified in the striatal lysates. Would the same be true for any compendium of neuronal genes?

3. Along those lines, are the 'novel' post-synaptic and other proteins identified in the striatum something that is unique to SNpc cells, or present in other subsets of neurons. The implication of involvement in PD seems overly speculative without real support.

4. Beyond the proteomic IDs, orthogonal methods of detecting some of the non-expected post-synaptic proteins in the striatum are mandatory for interpreting the validity of the IDs.

[Editors' note: further revisions were suggested prior to acceptance, as described below.]

Thank you for resubmitting your work entitled "Subcellular proteomics of dopamine neurons in the mouse brain reveals axonal enrichment of proteins encoded by Parkinson's disease-linked genes" for further consideration by *eLife*. Your revised article has been reviewed by 2 peer reviewers and the evaluation has been overseen by Gary Westbrook as the Senior Editor and a Reviewing Editor. The consensus discussion of the reviewers and editors is summarized below. We will look forward to hearing from you with a revised article and a response letter describing the changes made.

Essential revisions:

The manuscript has been greatly improved but all reviewers concur that there are some remaining issues that must be addressed. All reviewers applaud the quality and rigor of revised data and corresponding Figures 1-5. The reviewers and editors agree that the authors have carefully considered concerns and addressed the major questions surrounding methods and rationale for normalization of data as well as validation of key DA axon enriched proteins using orthogonal approaches. Further the flawed statistical analysis of axonal vs. somatodendritic enrichment for PD genes was removed.

However, a lingering remnant of the flawed enrichment dataset in PD is inappropriately held over (e.g., the title of the manuscript which must be revised). The reviewers and editors think that the enrichment strategy (Figure 6) continues to suffer from an imprecise GWAS list of genes that inaccurately infers a particular gene at a locus when the actual gene may not be known with the degree of precision required here. Further, the APEX2 strategy biases towards certain proteins and not others that are known to be important in PD but were excluded. For example, well-known PD-associated genes like LRRK2 and GBA might be excluded because the cytoplasmic APEX2 enzyme does not access many endolysosomal proteins, favoring instead distributed proteins with bias. APEX2 is not distributed in the soma in all compartments evenly, and may exclude important mitochondrial genes. The imprecision and bias in both approaches combined, yields a meaningless dataset that is included in Figure 6 and related Supplementary files. Even without these flaws, without comparator datasets, it is not clear whether there would be similar enrichments in any neuronal context, or what the non-neuronal proteome might be. Thus, Figure 6 and all related text referring to PD and PD enrichment that utilizes PD-linked genes should be removed, including references in the title and Supplementary files. The editors acknowledge that removal of these analyses may have an impact on authorship.

All reviewers felt that Figures 1-5 and related text were exciting and had sufficient impact on their own.

---

## [Author Response]

Essential revisions:1) All reviewers and the Reviewing editor agreed that additional protein validation is needed, potentially with orthogonal antibody staining approaches. While the synaptosome preps presented are supportive of the BioIDs, and may facilitate candidate selection, the synaptosome preps do not obviate the need for an orthogonal approach. In protein validation, the target could be evaluated in other circuits as well which would help provide insight into how unique the localization in the dopaminergic neurons really is compared to other types of neurons.

In the revised manuscript we have added antibody validation for several somatodendritic and axonal proteins identified by APEX2 proteomics, using both western blotting and immunohistochemistry (new Figure 4). We have also added a comparison of our mDA axonal proteomics data to a previously published dataset of cultured cortical axons (new Figure 4—figure supplement 1). Despite major differences in neuronal maturity, cell type, and methodology, we find that many features of our axonal proteome are conserved in cortical axons. However, we also identify axonal localization of proteins that appears specific to mature mDA neurons or to embryonic cortical neurons. Of particular interest, we use immunohistochemistry to confirm the axonal localization of Kv4.3 and GIRK2 in mDA neurons (new Figure 4), both of which are ion channels that had been presumed to be exclusively somatodendritic.

2) All reviewers and the Reviewing editor were confused by the steps taken by the authors to normalize the data. Two of the reviewer's thought that it is critical to normalize the abundance of biotinylated proteins from one lysate to the next to the amount of APEX2 enzyme present in the lysate, or provide a justification of why more peroxidase expression would not be expected to label more distinct protein substrates and label protein substrates to a higher level? Normalization methods of all mass spectrometry data should be included briefly in all relevant figure legends, as well as a clear work-flow in the Methods section.

We apologize for the confusion regarding the normalization of the mass spectrometry data. The Methods section “Proteomic Differential Expression Analysis and Filtering” explained that we used total intensity (i.e., the sum of all protein intensities within a given sample) to normalize protein abundances. We have expanded this section for added clarity (page 22):

“Total intensity normalized, log2 transformed protein abundances were used for visualization, clustering, and all differential abundance analyses. […] Differential abundance testing consisted of a Welch’s (unequal variance) t-test with Benjamini-Hochberg procedure to control the False discovery rate (FDR).”

and to assist readers, in the figure captions where we previously specified “log2(normalized intensity +1)”, we now write “log2(total intensity normalized abundance + 1)”. In addition, given the confusion about relative vs. absolute protein abundances, we have added further clarification to the Results section (page 6):

“To identify APEX2-dependent proteins captured by streptavidin pulldown, we normalized protein abundances to total protein intensity within each sample (see Methods) and directly compared APEX2^+^ to APEX2^-^ control samples (Figure 2e). […] We emphasize that the normalized proteomics data report on the *relative abundance* of proteins (relative to all protein captured by streptavidin pulldown in each sample), while the *absolute abundance* determined by western blot (Figure 2b) shows that most of the DA neuronal protein mass is axonal.”

Regarding normalization to the amount of APEX2, this concern is addressed in detail in several responses to reviewers below. Briefly, we find no evidence that the cytoplasmic abundance of APEX2 across mDA neuronal compartments is biased compared to other cytoplasmic proteins. Thus, axon vs. somatodendritic comparisons are unlikely to be affected by mouse-to-mouse variability in APEX2 expression, as the VM, MFB, and Str samples all come from the same mouse. The reviewers are correct that large differences in APEX2 expression (i.e., transduction efficiency of the AAV) between mice might be a concern for the extent of labeling. However, as we show in the manuscript, our biological replicates are highly reproducible in both the number of proteins identified and the correlation between within-group samples. Thus, viral expression is not a major source of variability between mice in our study. We found that the total intensity (i.e., the sum of all mapped peptides/proteins) was more variable between samples in the raw mass spectrometry data, likely due to technical variability in the LC-MS/MS data acquisition. As we show below, total intensity normalization and normalization to APEX2 produce virtually identical effect sizes (log2 fold changes), but total intensity normalization is much better at reducing within-group variability.

3) Two reviewers thought the comparative analysis of the PD GWAS data with the proteome IDs was imprecise at best, and potentially misleading. Some specific concerns include the conflation of strong recessive, dominant, and risk factor variants with potentially incorrect gene assignments, arbitrary inclusion of some genes and not others, overly relaxed false-discovery rates, and heavy implicit bias. All reviewers and Reviewing Editor agreed that the manuscript would be more focused without these experiments and the related claims about the heritable aspects of PD and PD-associated diseases.

We agree with the reviewers that the claims about heritable aspects of PD are unnecessary, and the points about grouping of recessive, dominant, and risk factor variants from GWAS for joint analysis are well-taken. Therefore, rather than conduct an underpowered analysis with each of these groups independently, we have chosen to remove the statistical analysis of axonal vs. somatodendritic enrichment for PD genes as a group (now Figure 6).

However, we disagree that the data should be removed from the manuscript. Assuming a protein has equal relative abundance in the somatodendritic and axonal cytoplasm, our protein abundance data suggest that ~90% of the total protein would then be axonal. It is therefore important to consider axonal localization in cell biology studies of PD-linked proteins, regardless of whether PD-linked proteins are statistically enriched in axons as a group compared to all neuronal proteins. Our dataset does not detect all PD-linked proteins, which could be due to the underlying cell biology (e.g., low expression levels in mDA neurons, such as LRRK2) or technical reasons (e.g., not detected by mass spec, or not accessible to cytoplasmic APEX2 labeling, such as GBA, which encodes a luminal lysosomal enzyme). Although we agree that these limitations preclude robust statistical analysis for PD-linked proteins as a group, we believe the localization pattern of the proteins we detect is of interest to the field, as it may provide insight on axonal proteins involved in PD neurodegeneration. We have modified the figure (now Figure 6c), the Results, and the Discussion to reflect these changes.

Reviewer #1 (Recommendations for the authors):The following suggestions for improvement are provided per figure for clarity, although some may be relevant more globally;Figure 1 is generally showing that the technique works in vivo and is good. One small question is with 1d, where it appears that the V5-APEX2 expression is higher in the striatum than in midbrain. Was this consistently seen and might it influence detection sensitivity later in the paper? It would be important to add a reference protein for loading to 1d and quantify relative expression in the two regions.

The western blot in Figure 1d is VM/Str tissue lysates first probed with streptavidin HRP and subsequently re-probed with anti-V5/APEX2. As noted in the manuscript, probing brain lysates or streptavidin pulldowns with streptavidin HRP produces strong bands in all lanes at ~75 and ~150 kDa, which are endogenously biotinylated proteins. These are labeled ‘ebp’ in Figure 1d and serve as an internal control that demonstrates equal protein loading. We note that the amount of V5-APEX2 in each sample is a function of:

1) the density of mDA neurons/axons in the tissue (VM vs. Str)

2) the relative abundance of APEX2 within the somatodendritic and axonal compartments of mDA neurons (VM vs. Str)

Even with equal protein loading, there is no expectation that V5-APEX2 levels should appear equal by tissue western blot, since mDA cell bodies/dendrites vs. axons do not comprise an equal fraction of total protein for VM and striatal tissue, respectively. Given the NES tag, we expect V5-APEX2 to be evenly distributed throughout the axonal and somatodendritic cytoplasm. The western blot in Figure 1d likely reflects a higher fractional protein contribution from mDA neurons in the striatum compared to the VM, and cannot quantify relative V5-APEX2 expression within the mDA neuronal cytoplasm. Instead, we analyzed the fluorescence intensity of TH, tdTomato, and V5-APEX immunostaining in confocal images of DAT^IRES-Cre^/Ai9^tdTomato^ mice injected with AAV5-CAG-DIO-APEX2-NES.

As shown in the new panels Figure 1—figure supplement 1c-d, we found no evidence that the concentration of APEX2 is higher in mDA axons than in soma and dendrites: rather, it appears that the concentration of APEX2, TH, and tdTomato may be slightly higher in the soma. We now report in the Results (page 4):

“All three markers displayed intense staining throughout the DA neuronal cytoplasm, including dendrites in the VM and axonal projections in the striatum

(Figure 1b). […] Thus, injection of Cre-dependent AAV-APEX2NES (hereafter referred to as APEX2) into the VM of DAT^IRES-Cre^ mice leads to robust expression of APEX2 throughout the DA neuronal cytoplasm."

Figure 2 is the main results figure and again largely makes sense. Here, I would again like to see an evaluation of the amount of V5-APEX2 in each region relative to control, so we can understand the apparent discrepancy between number of proteins detected and % biotinylation between ventral midbrain and striatum. It is notable here that the control is different from figure 1, which was -H_2_O_2_, vs no APEX2. It would therefore be important to include validation experiments using immunoblotting and also include the controls from figure 1 to be sure that these are dependent on APEX2 activity.

We respectfully point out that the ‘amount of V5-APEX2 in each region relative to control’ is a misunderstanding, as the controls in these experiments do not have APEX2. The discrepancy between number of proteins detected and % biotinylation is expected due to the *greater total abundance* of mDA neuronal cytoplasm, and thus V5-APEX2, in the striatum. As demonstrated above, these differences do not reflect a higher relative abundance of V5-APEX2 within mDA axons, but rather that majority of the total mDA neuronal volume is axonal.

Regarding the controls, as established by the Ting Lab (Hung et al., Nat Protocols 2016): “Negative controls are performed alongside, with APEX2 omitted, BP omitted or H_2_O_2_ omitted.” We have used all three controls in our work. In Figure 1, we show that in the presence of APEX2, omission of H_2_O_2_ or BP leads to a complete absence of biotinylation. In Figure 2b, we show that biotinylation is undetectable in the absence of APEX2 (aside from endogenously biotinylated proteins). However, it is possible that very low levels of endogenous peroxidases could catalyze BP and H_2_O_2_-dependent labeling in the absence of APEX2. The omission of H_2_O_2_ would not capture such low level labeling, and H_2_O_2_ is also likely to have biological effects on slices that could potentially alter non-specific binding to the streptavidin beads. Therefore, we feel that for proteomics experiments, omission of APEX2 is the best control for endogenous peroxidase activity and non-specific binding, rather than omission of H_2_O_2_.

Figure 3 is generally fine apart from the discrepancy in FDR p values selected for cutoffs. FDR<0.15 is too liberal, especially given that log fold cutoffs appear not to have been applied and that t-tests were used without evaluation of normal distributions that would be difficult from low n of samples. Again some validation of key results is needed.

We apologize for the confusion: the GO analysis was conducted only on APEX2-enriched proteins that met the criteria of FDR < 0.05 for VM vs. Str. As shown in the center of Figure 3, there are two shades of red/blue used to color proteins based on the significance of VM vs. Str enrichment: a darker shade for those meeting FDR < 0.05 (121/149, 81% of proteins shown), and another for those meeting FDR < 0.15 (12/149, 8% of proteins shown). Proteins present in both compartments at roughly equal levels are shown in gray (16/149, 11% of proteins shown). The goal was to include a more complete representation of detected proteins relevant to the Ontologies shown. As an example, the glycolytic enzymes HK1 and GAPDH were detected in all APEX2 samples but were not differentially abundant in the VM vs. Str comparison. However, 7/9 of the other glycolytic enzymes were found to be enriched in Str > VM (shown in dark red) and thus Glycolysis is highlighted as a Str-enriched Gene Ontology. Rather than omit HK1 and GAPDH from the figure, we show these proteins in grey to indicate that they are detected in both compartments, but not differentially abundant.

To highlight the validity of our Str vs. VM comparisons using an orthogonal approach, we measured the log2 fold change (Str/VM) for several proteins using western blotting. As shown in new Figure 4a-b, the effect sizes determined by western blot (absolute abundance) and proteomics (relative abundance) are well correlated.

Figure 4 particularly needs validation of proteins that are expected to be post-synaptic. I am sure that the literature distinctions between pre and post synaptic are less rigid than might be inferred, but some evaluation of accuracy of this separation is needed.

The previous Figure 4 (now Figure 5) did not show proteins that are expected to be post-synaptic, and we surmise that the reviewer is referring to previous Figure 4—figure supplement 1 (now Figure 5—figure supplement 1). As noted in the discussion, distinguishing the ultrastructural localization of these broadly expressed proteins would require immunoelectron microscopy and suitable antibodies, and the often poor binding specificity of most antibody reagents is a major concern for reproducibility, especially in brain IHC studies (PMID 16885215). Indeed, we found that many of the antibodies we purchased produced non-specific IHC staining and numerous bands on western blot that precluded identification of the correct protein (not shown). These experiments often require knockout cells/mice to be properly interpreted.

For some proteins we identified, such as Syt17, visualization of protein localization requires expression of tagged fusion proteins (Ruhl et al., 2019, PMID 31387992). In this case, we found orthogonal evidence of Syt17 in mDA neurons via the GENSAT project mouse that expresses a Syt17-EGFP fusion protein. EGFP staining of these mice showed intense labeling of mDA neurons in the midbrain as well as their axons in the MFB and striatum (new Figure 6—figure supplement 3).

While recognizing the limitations with antibodies, we found that knockout-validated antibodies for Kv4.3 and GIRK2 were suitable for high resolution confocal immunohistochemistry (new Figure 4d-e). These ion channels are known to be highly expressed in the somatodendritic compartment of mDA neurons, but previous work suggested that GIRK2 was not present in mDA axons (Martel et al., 2011, PMID 21647367). Our APEX2 data showed that while were Kv4.3 and GIRK2 were higher in VM vs. Str samples, both proteins were present in mDA axons. Consistent with the APEX2 data, we demonstrate that immunoreactivity for Kv4.3 and GIRK2 co-localized with mDA axons within the MFB and the striatal neuropil (new Figure 4d-e). As noted in the immunostaining and comparison to cortical axon proteomics, the axonal localization of these ion channels in mDA neurons is not shared with cortical neurons. These data support our finding that some, but not all, canonical somatodendritic proteins are localized to mDA axons. Indeed, consistent with the APEX2 data, we show that Homer2, an mDA neuron-enriched post-synaptic protein, is prominently localized within dopaminergic dendrites in the SNr but not within MFB/Str axons (new Figure 4—figure supplement 2).

Figure 5 is the figure that has most problems. Figure 5a is a schematic and 5b tells us that there is partial agreement between proteome and scRNA-seq, which is to be expected. But, 5c, has little informational quality for multiple reasons. For GWAS SNPS, the selection of nearest gene to lead SNP is only true in some uncertain proportion of loci so whether INPP5F is the gene at the Chr10 locus that includes BAG3 and RGS10 is impossible to evaluate. For the smaller set of Mendelian loci, whether we should combine PD and atypical and dystonia is hard to evaluate. At the same time, there are multiple loci that are not in this set – LRRK2 and GBA being very obvious. So at best, this dataset says that some PD genes are in dopamine neurons, which is unsurprising, but not all PD genes are dopamine neuronal. The real problem here is in decision of numerator and denominator. For other tools more widely used in GWAS (FUMA, MAGMA etc) the test set is all candidate genes vs all expressed in a given cell type. Here, the authors compare a set they detect in the cells and then look for cell body vs axon enrichment, which is fundamentally less precise or informative. Authors should look for enrichment within each dataset for all Mendelian PD genes or all GWAs hits, which should include all reasonable candidates within LD-defined bounds of each locus. Even if this turns out to be more than chance, the authors must discuss limitations – DJ-1 is found here but very notably not PINK1/parkin. Such patterns might easily be explained by chance ordering of proteins in Str vs VM and should be discussed adequately.

We agree on many of these points, particularly the grouping of Mendelian PD, atypical PD, dystonia, and GWAS genes for a joint analysis. We have chosen to remove the statistical analysis and claims about axonal enrichment of PD-linked proteins as a group. However, we believe that the localization pattern for the proteins we do detect is worth reporting. We agree about the GWAS SNPs and we have expanded the heatmap (now Figure 6c) to include all genes within the PD GWAS risk loci as defined by Nalls et al. (2019). To acknowledge the limitations of our dataset, we have also edited the Results (page 13):

“Nonetheless, we are limited to analysis of proteins detected by our APEX2 proteomics, which does not detect all PD-linked proteins. […] Thus, while our dataset highlights subcellular localizations for follow-up studies, it does not establish the absence of PD-linked proteins at a given site.”

Reviewer #2 (Recommendations for the authors):1. When doing streptavidin-dependent pulldowns of biotinylated proteins the authors write 'immunoprecipitation'. This is not correct since no antibodies are used. The authors should be technically correct and talk about 'pull-downs' or 'streptavidin-dependent purification of biotinylated proteins.' I realize most readers will understand what they are trying to say, but for such a technically excellent paper I think the authors shouldn't use wrong terminology.

We agree and regret the error. We have changed all instances of ‘immunoprecipitation / IP’ to ‘streptavidin pulldown’.

2. In some instances I found the figures to be overly 'busy'. For example, Figure 3d is very busy with a great deal of speculation included about protein function. No studies here actually test biology of the candidates identified by mass spectrometry. The faith (and I label it as such) placed in GO analysis by the authors is not justified. On the other hand, I applaud the authors for attempting to place some of the identified proteins in biological context. What I'd prefer to see is a bit of a disclaimer about the robustness of GO analysis and also include more statements about how this is a discovery approach that will require many follow-up studies to elucidate protein function.

We agree that our proteomic profiling is a discovery approach, and in Figure 3d we intended to highlight the breadth and depth of proteins identified in our mass spec data. The sets of proteins displayed were chosen on the basis of the axonal vs. somatodendritic GO analysis. We agree that GO analysis does not establish biological function, but rather highlights areas for further follow-up studies. We have modified the Results (page 9):

“While our GO analysis is broadly consistent with established features of neuronal polarization (e.g., somatodendritic enrichment of terms such as ‘Postsynapse’ and ‘RNA binding’), findings from this discovery approach will require follow-up studies to elucidate compartment-specific function of the identified proteins.”

Reviewer #3 (Recommendations for the authors):The compendium seems rigorous and potentially useful to the field in understanding TH neurons in the SNpc.1. I do not understand why biotinlyated protein abundance in the different preparations from the different compartments are not normalized to APEX2 abundance in the relative quantifications. Overall, the process of normalization was unclear, including how differing AAV transduction efficiency is factored into calculated biotinylated proteins. Unless i am fundamentally mistaken, APEX2 labeling occurs so quickly that the relative abundance of enzyme will influence not only the number of proteins that the mass spectrometry analysis can identify, but also the relative abundance when comparing one preparation of protein to the next.

We minimize mouse-to-mouse variability by conducting all surgeries for a cohort of mice in a single day with the same aliquot of virus, and conducting the slicing/labeling procedures on at least 2 mice per day in balanced pairs of control/APEX2. We minimize protein preparation variability by freezing the labeled tissue at -80C until all samples in a batch are ready for protein extraction and streptavidin pulldown. We then process all samples at the same time, including streptavidin digest and mass spectrometry. The major proteomics dataset in this paper is derived from a single cohort of mice, and the extent of biotinylation measured by streptavidin HRP western blotting was comparable between mice (Figure 2b). Overall, we show that our workflow facilitates strong reproducibility in our APEX2 mass spectrometry data, both for the number of peptides/proteins detected per sample (Figure 2c) and the correlation of biological replicates (Figure 2—figure supplement 1c).

The reviewer’s suggestion to normalize protein abundances to APEX2 abundance within each sample is an interesting one, but a key disadvantage of that approach is that it eliminates the possibility of comparing APEX2^+^ samples to our APEX2^-^ controls. Another recent proximity labeling study suggested that normalization to the endogenously biotinylated protein, propionyl-CoA carboxylase alpha chain (Pcca), reduced variability between different batches of APEX samples (Frankenfield et al., 2020; PMID 33201688).In our study we normalized protein abundances to the total intensity within each sample. To compare total intensity normalization to APEX2 or Pcca normalization, we compared the intragroup coefficient of variance (CV), a commonly employed metric in evaluating proteomic normalization methods (Välikangas et al., 2018; PMID 27694351). Compared to total intensity, APEX2 or Pcca normalization both increased the intragroup variability for the majority of proteins (Author response image 1).

**Author response image 1. sa2fig1:** Comparison of proteomic normalization methods. (A-B) Intragroup coefficient of variation (CV) for APEX2^+^ samples after normalization to total intensity vs. Pcca or APEX2 abundance. (C-D) Log2 Fold Changes and FDR q-values for VM vs. Str APEX2+ comparisons after normalization to total intensity or APEX2 abundance.

We also re-analyzed VM vs. Str samples to determine whether normalization to APEX2 would alter the axonal vs. somatodendritic enrichment profile. As shown in Author response image 1, the log2 Fold Changes are virtually identical for both normalization methods. However, for the majority of proteins the FDR q-values are much higher (less significant) with APEX2 normalization (Author response image 1), consistent with the increased intragroup variability. These results demonstrate that normalization to total protein intensity reduces within-group variability better than normalization APEX2 or Pcca. However, they do not rule out the possibility that normalization to Pcca or APEX2 would be useful in future studies containing multiple batches of samples, given that our study was rigorously designed to mitigate experimental batch effects.

2. Along those lines, since there are simply more proteins IDed in the striatum, with a larger 'interactome' afforded by more substrate material, and with higher possible proportional abundance of APEX2 , it makes sense that there are more PD- GWAS genes identified in the striatal lysates. Would the same be true for any compendium of neuronal genes?

The reviewer is correct that there is more total substrate material in the striatum due to the massive mDA axons, but not a higher relative abundance of APEX2 in axons (see above). The point is well taken that the deeper coverage of striatal samples might bias which proteins are detected; we attempted to mitigate this bias by analyzing only PD-GWAS genes that were detected in both VM and Str samples. Nonetheless, as described above and in response to the reviews and Editor’s points, we have chosen to omit this analysis and remove claims about PD genes as a group.

3. Along those lines, are the 'novel' post-synaptic and other proteins identified in the striatum something that is unique to SNpc cells, or present in other subsets of neurons. The implication of involvement in PD seems overly speculative without real support.

We respectfully note that we did not suggest involvement of any post-synaptic proteins in PD, and that similar axonal datasets from mature neurons in vivo do not exist yet. However, as shown in new Figure 5—figure supplement 1c-d, many of the post-synaptic proteins we detected in the striatum are also present in cultured cortical axons.

4. Beyond the proteomic IDs, orthogonal methods of detecting some of the non-expected post-synaptic proteins in the striatum are mandatory for interpreting the validity of the IDs.

In this revision we have now confirmed the axonal localization of the potassium channels GIRK2 (*Kcnj6*) and Kv4.3 (*Kcnd3*) using immunohistochemistry (new Figure 4). These data highlight the ability of APEX2 labeling to reveal low abundance axonal proteins often assumed to be localized only in the somatodendritic compartment of neurons.

[Editors' note: further revisions were suggested prior to acceptance, as described below.]

Essential revisions:The manuscript has been greatly improved but all reviewers concur that there are some remaining issues that must be addressed. All reviewers applaud the quality and rigor of revised data and corresponding Figures 1-5. The reviewers and editors agree that the authors have carefully considered concerns and addressed the major questions surrounding methods and rationale for normalization of data as well as validation of key DA axon enriched proteins using orthogonal approaches. Further the flawed statistical analysis of axonal vs. somatodendritic enrichment for PD genes was removed.However, a lingering remnant of the flawed enrichment dataset in PD is inappropriately held over (e.g., the title of the manuscript which must be revised). The reviewers and editors think that the enrichment strategy (Figure 6) continues to suffer from an imprecise GWAS list of genes that inaccurately infers a particular gene at a locus when the actual gene may not be known with the degree of precision required here. Further, the APEX2 strategy biases towards certain proteins and not others that are known to be important in PD but were excluded. For example, well-known PD-associated genes like LRRK2 and GBA might be excluded because the cytoplasmic APEX2 enzyme does not access many endolysosomal proteins, favoring instead distributed proteins with bias. APEX2 is not distributed in the soma in all compartments evenly, and may exclude important mitochondrial genes. The imprecision and bias in both approaches combined, yields a meaningless dataset that is included in Figure 6 and related Supplementary files. Even without these flaws, without comparator datasets, it is not clear whether there would be similar enrichments in any neuronal context, or what the non-neuronal proteome might be. Thus, Figure 6 and all related text referring to PD and PD enrichment that utilizes PD-linked genes should be removed, including references in the title and Supplementary files. The editors acknowledge that removal of these analyses may have an impact on authorship.All reviewers felt that Figures 1-5 and related text were exciting and had sufficient impact on their own.

We have made the following revisions:

1. We removed any mention of Parkinson’s disease from the title.

2. We removed all text referring to the localization of proteins encoded by the Parkinson’s disease-linked genes shown in Figure 6c.

3. We removed the components of Figure 6 that described our analysis of Parkinson’s disease-linked genes. The essential revisions required the complete removal of Figure 6, but both the editorial and reviewer comments seem to refer only to the component of Figure 6 that is related to Parkinson’s disease, which would include part of the schematic in Figure 6a and the heatmap in Figure 6c. However, we did not remove Figure 6b, where we present an analysis of localization of dopamine neuron markers identified from singlecell RNA-seq. This component of Figure 6 is unrelated to Parkinson’s disease, and there is no mention of this analysis in the editorial or reviewer comments.

4. We removed all supplementary figures and source data related to the localization of proteins encoded by Parkinson’s disease-linked genes shown in Figure 6c.